# Defending Against Adversarial Attacks via Neural Dynamic System

**Xiyuan Li**
School of Computer Science
Wuhan University
Lee_xiyuan@outlook.com

**Xin Zou**
School of Computer Science
Wuhan University
zouxin2021@gmail.com

**Weiwei Liu**[*]
School of Computer Science
Wuhan University
liuweiwei863@gmail.com

## Abstract

Although deep neural networks (DNN) have achieved great success, their applications in safety-critical areas are hindered due to their vulnerability to adversarial attacks. Some recent works have accordingly proposed to enhance the robustness of DNN from a dynamic system perspective. Following this line of inquiry, and inspired by the asymptotic stability of the general nonautonomous dynamical system, we propose to make each clean instance be the asymptotically stable equilibrium points of a slowly time-varying system in order to defend against adversarial attacks. We present a theoretical guarantee that if a clean instance is an asymptotically stable equilibrium point and the adversarial instance is in the neighborhood of this point, the asymptotic stability will reduce the adversarial noise to bring the adversarial instance close to the clean instance. Motivated by our theoretical results, we go on to propose a nonautonomous neural ordinary differential equation (ASODE) and place constraints on its corresponding linear time-variant system to make all clean instances act as its asymptotically stable equilibrium points. Our analysis suggests that the constraints can be converted to regularizers in implementation. The experimental results show that ASODE improves robustness against adversarial attacks and outperforms the state-of-the-art methods.

## 1 Introduction

Deep neural networks (DNNs) have achieved great success in a range of tasks, including image recognition [1, 2], scene segmentation [3, 4, 5] and action recognition [6, 7]. However, their performance can be significantly affected by human-imperceptible perturbations that can drastically change the network's output [8, 9]. This phenomenon seriously restricts the application of DNN in safety-critical fields such as automatic driving.

Recently, some methods have been proposed to defend against adversarial attacks from the perspective of dynamic systems. The paper [10] proposes a time-invariant steady neural ODE (TisODE) to limit the evolution of the curves by forcing the integrand to be close to zero. However, this approach does not guarantee that small perturbations of the initial point will lead to small perturbations in the output at time $T$. To reduce perturbations on the initial point, SODEF [11] causes the extracted features to be located within a neighborhood of the Lyapunov-stable equilibrium points of the autonomous ODE. However, there are still some drawbacks of SODEF: 1. In the papers [1, 12, 13], the general dynamical system explicitly depends on the argument $t$ and is referred to as a nonautonomous dynamical system. However, SODEF only considers the autonomous case, which is simply a special class of ODE. 2. As shown in Figure 1 (a), Lyapunov stability only controls the perturbations of the input rather than eliminating the effects of perturbations; simply maintaining the perturbations may still lead

---

[*]Corresponding author

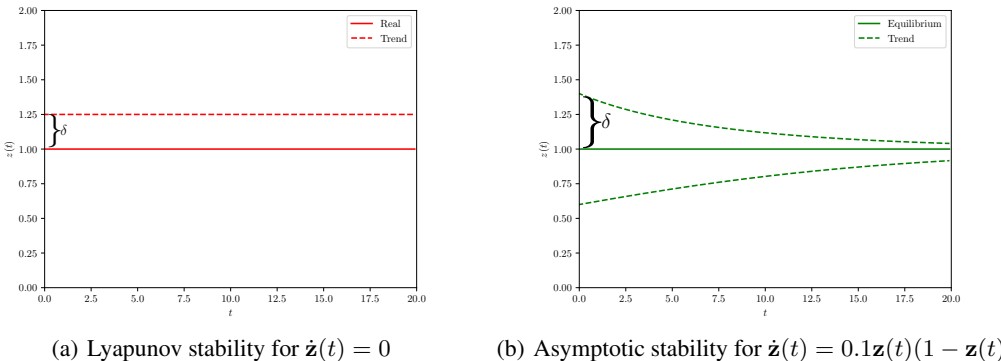

(a) Lyapunov stability for $\dot{\mathbf{z}}(t) = 0$  (b) Asymptotic stability for $\dot{\mathbf{z}}(t) = 0.1\mathbf{z}(t)(1 - \mathbf{z}(t))$

Figure 1: Comparison of Lyapunov stability and asymptotic stability at equilibrium point $\mathbf{z} = 1$.

to the misclassification of SODEF. 3. SODEF ensures that the output of the extractor is located in the neighborhood of the Lyapunov-stable equilibrium points; thus, the performance of SODEF depends on the extractor. Specifically, if the extracted features deviate from real features, SODEF would regard the false features as the Lyapunov-stable equilibrium points; accordingly, the perturbed input of neural ODE will converge to the incorrect features, which would lead to misclassification of SODEF. These are also the major difference between SODEF and our method.

To solve the problems, in this paper, we first divide the instances into clean instance $\mathbf{x}$, perturbed instance $\widetilde{\mathbf{x}}$, and contaminated instance $\hat{\mathbf{x}}$, which are shown in Figure 2. Instances in the yellow and blue region of the same ball naturally have the same label $\mathbf{y}$. Then, the vulnerability of DNN to perturbation can be illustrated as following: The function $f$ is constructed by DNN. $f$ makes correct prediction when the input instance is clean, but makes wrong prediction when the input is contaminated. In other words, we have $f(\mathbf{x}) = \mathbf{y}$ and $f(\hat{\mathbf{x}}) \neq \mathbf{y}$. Therefore, we make $\hat{\mathbf{x}}$ converge to $\mathbf{x}$ by dynamic system to improve the robustness of DNN. Compared with SODEF, we consider a more general nonautonomous case:

$$\frac{d\mathbf{z}(t)}{dt} = \mathbf{h}(\mathbf{z}(t), t). \tag{1}$$

As shown in Figure 1 (b), asymptotic stability helps to reduce the perturbation as $t$ increases. Therefore, we make each clean instance into an asymptotically stable equilibrium point and locate the perturbed instance in its neighborhood. In order to do this, we transform the nonautonomous

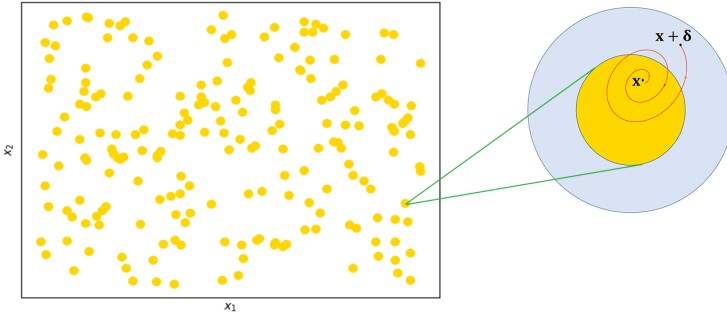

Figure 2: The set including all yellow balls in the coordinate space is the clean instance domain $\mathcal{X}$. For convenience, we magnify a yellow ball on the right. The point in the yellow ball corresponds to clean instance $\mathbf{x}$. We place perturbations on the instance in the yellow ball to form a perturbed instance $\widetilde{\mathbf{x}}$ that is still recognizable. The perturbed instance $\widetilde{\mathbf{x}}$ corresponds to the blue and yellow regions. The instance in the blue region is referred to as the contaminated instance $\hat{\mathbf{x}}$. Instances in the yellow and blue region of the same ball naturally have the same label $\mathbf{y}$. The red spiral line is the trajectory along which the contaminated instance $\hat{\mathbf{x}}$ converges to clean instance $\mathbf{x}$.

ODE (1) into a time-variant linear state-space equation, as follows:

$$\frac{d\mathbf{z}(t)}{dt} = \mathbf{A}(t)(\mathbf{z}(t) - \mathbf{x}). \tag{2}$$

According to nonlinear dynamic system theory [14], (2) is exponentially stable if the nonautonomous ODE (1) is a slowly varying system and the Jacobian matrix $\mathbf{A}$ to which each frozen system $\dot{\mathbf{z}}(t) = \mathbf{h}(\mathbf{z}(t), \tau)$ corresponds is exponentially stable. This holds because if all eigenvalues of $\mathbf{A}$ have negative real parts, $\mathbf{A}$ is exponentially stable. We use the Levy-Desplanques theorem [15] to add constraints to the elements of Jacobian matrix $\mathbf{A}$ to ensure that all eigenvalues of $\mathbf{A}$ have negative real parts. We solve the stability of (1) by imposing constraints on its linearization. Additionally, rather than applying neural ODE to the output of the extractor, we instead apply neural ODE directly on the instance $\mathbf{x} \in \mathcal{X}$ to eliminate the effects of perturbations. The process is also illustrated in Figure 2, where the perturbed instance $\mathbf{x} + \delta$ converges to the clean instance $\mathbf{x}$ along the red trajectory. Based on the theoretical analysis above, we propose a stable neural ODE (ASODE) with asymptotic stable equilibrium points for defending against adversarial attacks, and further construct an optimization formulation to make it satisfy the asymptotic stability. Moreover, we conduct experiments to compare the robustness of ASODE with ODE-Net [12], TisODE-Net [10] and SODEF [11] under different adversaril attacks on the CIFAR-10 [16] and MNIST [17] datasets. The results shows that ASODE outperforms the state-of-the-art SODEF. Moreover, both ASODE and SODEF are far more robust than ODE-Net and TisODE-Net. For example, compared with the state-of-the-art neural ODE network SODEF under PGD attack on the MNIST and CIFAR-10 datasets, our ASODE improves the classification accuracy of adversarial examples by $1.60\%$ and $1.76\%$, respectively. Furthermore, compared with the TisODE-Net on the same datasets under PGD attack, ASODE improves the classification accuracy of adversarial examples by $45.07\%$ and $53.53\%$, respectively.

## 2  Related Work

**Neural ODEs** The relationship between the ODEs and neural networks has been illustrated by [12, 18]. The papers [19, 20, 21] regard residual networks as a form of explicit Euler discretization with a unit step size. Based on this, various perspectives and methods from numerical analysis have been employed to improve the network architecture [18, 22], minimize memory overload [12], reduce training time [23], and facilitate adaptivity to other models (e.g., transformers [24]).

**Adversarial Defense** Many methods have been proposed to improve the robustness of neural networks via training using different strategies, including Bayesian adversarial learning [25], various regularization [26, 27], adversarial training and its variants [28, 29, 30, 31]. Moreover, some works explore adversarial robustness from the perspective of the stability of the numerical ODE or dynamic system. For example, [32] reduces feature noise using ODE to develop a robust architecture. [13] proposes a new robust architecture, improving the robustness of the original residual network family via the implicit Euler method of ODE.

Among the works, [33] is the first work to use control theory and dynamic systems to improve robustness, which is different from our work in the types of nonlinear system and the inputs. [34] uses the property of radial basis function (RBF) to improve the robust, not ODE. In the work most closely related to ours, [11] designs a stable neural autonomous ODE with Lyapunov-stable equilibrium points for defending against adversarial attacks. The Lyapunov-stability of the autonomous ODE ensures that the input features from the extractor with a small perturbation converge to the unperturbed input features. We study a more general case called a nonautonomous system. Based on this, we directly reduce the adversarial noise on the instance by ensuring that the perturbed instances are in the neighborhood of asymptotically stable equilibrium points of nonautonomous ODE. Therefore, we purify the sample and improve the network robustness.

## 3  Preliminaries

Let $\mathcal{D}$ be a probability distribution over $\mathcal{X} \times \mathcal{Y}$, where $\mathcal{X}$ is the instance domain set and $\mathcal{Y}$ is the set of labels. $(\mathbf{x}_n, \mathbf{y}_n)$ is sampled from $\mathcal{D}$, where $\mathbf{x}_n \in \mathbb{R}^d$, $\mathbf{y}_n \in \mathbb{R}^m$ and $n = 1, 2, \cdots, N$. $\mathcal{D}_x$ is the marginal distribution of $\mathbf{x}$ over $y$. $h(\mathbf{z}(t), t)$ is a continuous function of $\mathbb{R}^d \times \mathbb{R}^+$ into $\mathbb{R}^d$; here, $t \in \mathbb{R}^+$, $\mathbb{R}^+ = \{t \in \mathbb{R} : t \geq 0\}$, $\mathbf{z} \in \mathbb{R}^d$, $\mathbf{z}(t) : \mathbb{R}^+ \to \mathbb{R}^d$. Moreover, we use $\nabla h(\mathbf{z}, t)$ to denote its Jacobian at $\mathbf{z}$. $C^1$ represents the function with first-order derivative; $|| \cdot ||$ denotes

the Euclidean norm; $\mathbf{I}$ is a $d$-dimensional vector in which each element is equal to 1. Finally, $B_\delta(\mathbf{x}^*) = \{\mathbf{x} \in \mathbb{R}^d : ||\mathbf{x} - \mathbf{x}^*|| < \delta\}$.

The $d$-dimensional instance $\mathbf{x} \in \mathcal{X}$ can be seen as a point in the space $\mathbb{R}^d$. However, as shown in Figure 3 (c), not every point of $\mathbb{R}^d$ has the label $\mathbf{y}$. Figure 3 (b) is the perturbed Figure 3 (a), in which the perturbation can be arbitrarily small. Figure 3 (b) naturally has the label "6" and is in the instance domain. For each $\mathbf{x} \in \mathcal{X}$, there exists a continuous bounded neighborhood $B_\delta(\mathbf{x})$ such that $B_\delta(\mathbf{x}) \subset \mathcal{X}$ and the instances in $B_\delta(\mathbf{x})$ have the same label $\mathbf{y}$. Thus, the instance domain $\mathcal{X}$ corresponds to the bounded ball cloud in $\mathbb{R}^d$, which is represented by the yellow balls in the coordinate space of Figure 2. In addition, if we apply perturbation to each $\mathbf{x} \in \mathcal{X}$ to form a new instance domain $\widetilde{\mathcal{X}}$ the elements of which are denoted by $\widetilde{\mathbf{x}}$, then we can determine that $\widetilde{\mathcal{X}}$ includes the adversarial samples and $\mathcal{X} \subset \widetilde{\mathcal{X}}$. We make $\mathcal{D}_{\widetilde{\mathbf{x}}}$ the marginal distribution of $\widetilde{\mathbf{x}}$ over $\mathbf{y}$. These conclusions are directly demonstrated in the magnified ball of Figure 2. Based on the analysis of this magnified ball, we define a clean instance for $\mathbf{x} \in \mathcal{X}$, perturbed instance for $\widetilde{\mathbf{x}} \in \widetilde{\mathcal{X}}$, and contaminated instance for $\hat{\mathbf{x}} \in \{\widetilde{\mathbf{x}} : \widetilde{\mathbf{x}} \in \widetilde{\mathcal{X}}, \widetilde{\mathbf{x}} \notin \mathcal{X}\}$.



| (a) Clean instance | (b) Perturbed instance | (c) No label instance |

Figure 3: Instances sampled from $\mathbb{R}^{28 \times 28}$. The perturbed instance is in the neighborhood of the clean instance, but no label instance is far away from any clean instance.

Consider a nonautonomous initial value ODE problem,

$$\begin{cases} \dfrac{d\mathbf{z}(t)}{dt} = \mathbf{h}(\mathbf{z}(t), t), & t \geq t_0, \\ \mathbf{z}(t_0) = \mathbf{z}_0 \end{cases} \quad (3)$$

Suppose the function $\mathbf{h}$ satisfies a global Lipschitz condition [14, section 2.4.2], then, the solution of (3) is exist and unique over $t \in [0, \infty)$. The solution of (1) is denoted as $\mathbf{z}(t)$ with input $\mathbf{z}(0)$ and output $\mathbf{z}(T)$. In (3), $\mathbf{z}_0$ is the initial value of $\mathbf{z}(t)$ at $t = t_0$. If $t = t_0 = 0$, we have $\mathbf{z}(0) = \mathbf{z}_0 = \mathbf{x}$. Meanwhile, let $\mathbf{s}(\mathbf{z}_0, t_0, t)$ denote the solution of (3) corresponding to the initial input $\mathbf{z}_0$ at $t_0$. For simplicity, we use $\mathbf{s}(\mathbf{x}, t)$ to denote $\mathbf{s}(\mathbf{z}_0, 0, t)$. Obviously, if the initial value for ODE (1) is given by $\mathbf{z}(0) = \mathbf{x}$, the two representations are equivalent, and we have $\mathbf{z}(t) = \mathbf{s}(\mathbf{x}, t)$, $\mathbf{z}(0) = \mathbf{s}(\mathbf{x}, 0) = \mathbf{x}$, $\mathbf{z}(T) = \mathbf{s}(\mathbf{x}, T)$.

**Definition 1 (Equilibrium [14])** *A vector $\mathbf{x}^*$ is said to be an equilibrium of (1) if, $\mathbf{h}(\mathbf{x}^*, t) = 0$, $\forall t \geq 0$.*

According to Definition 1, for any given $\mathbf{x} \in \mathcal{X}$, if the solution of (3) is $\mathbf{z}(t) = \mathbf{x}, t \geq 0$, then $\mathbf{h}(\mathbf{x}, t) = \frac{d\mathbf{x}}{dt} = 0$, and $\mathbf{x}$ is an equilibrium of (1). On the other hand, if $\mathbf{x}$ is an equilibrium, (1) has the unique solution $\mathbf{z}(t) = \mathbf{x}, t \geq 0$ for the initial value $\mathbf{z}(0) = \mathbf{x}$. In other words, if a system starts in an equilibrium, it remains in that state thereafter.

**Definition 2 (Stability [35])** *A constant vector $\mathbf{x}^* \in \mathbb{R}^d$ is a stable equilibrium point for (1) if, for each $\epsilon > 0$ and each $t_0 \in \mathbb{R}^+$, there exists $\delta(\epsilon, t_0)$ such that for each $\mathbf{z}_0 \in B_\delta(\mathbf{x}^*)$, $||\mathbf{s}(\mathbf{z}_0, t_0, t) - \mathbf{x}^*|| < \epsilon, \forall t \geq t_0$.*

As shown in Figure 1 (a), whose ODE is $\dot{\mathbf{z}}(t) = 0$ with $\mathbf{z}(0) = \mathbf{x} \in \mathcal{X}$ (here, $\mathbf{x}$=1), for each $\epsilon > 0$, there exists $\delta < \frac{\epsilon}{d}\mathbf{I}$ such that $||\mathbf{s}(\mathbf{x} + \delta, t) - \mathbf{x}|| = ||\mathbf{x} + \delta - \mathbf{x}|| = ||\mathbf{x} + \frac{\epsilon}{d}\mathbf{I} - \mathbf{x}|| < \epsilon$. Therefore, $\mathbf{x} \in \mathcal{X}$ is the stable equilibrium point of ODE (1) and also the Lyapunov-stable equilibrium point.

**Definition 3 (Attractivity [35])** *A constant vector $\mathbf{x}^* \in \mathbb{R}^d$ is an attractive equilibrium point for (1) if for each $t_0 \in \mathbb{R}^+$, there exists $\delta(t_0) > 0$ such that for each $\mathbf{z}_0 \in B_\delta(\mathbf{x}^*)$, $\lim\limits_{t \to +\infty} ||\mathbf{s}(\mathbf{z}_0, t_0, t) - \mathbf{x}^*|| = 0$.*

**Definition 4 (Asymptotic stability [35])** *A constant vector $\mathbf{x}^* \in \mathbb{R}^d$ is asymptotically stable if it is both stable and attractive.*

It is worth noting that if an equilibrium is exponentially stable, it is also asymptotically stable with exponential convergence. As shown in Figure 1, $\mathbf{x} = 1$ in Figure 1 (b) is an attractive equilibrium point, but $\mathbf{x} = 1$ in Figure 1 (a) is not. Moreover, $\mathbf{x} = 1$ is also asymptotically stable in Figure 1 (b).

Based on the notation above, we aim to make the contaminated instance $\hat{\mathbf{x}}$ converge to the clean instance $\mathbf{x}$. In order to accomplish this evolution, we impose constraints on the ODE (1) to output $\mathbf{z}(T) = \mathbf{x}$ when the input is $\mathbf{z}(0) = \hat{\mathbf{x}} \in B_\delta(\mathbf{x})$. In order to let $\lim_{t \to +\infty} ||\mathbf{s}(\hat{\mathbf{x}}, t) - \mathbf{x}|| = 0$, where $\hat{\mathbf{x}} \in B_\delta(\mathbf{x})$, we make all $\mathbf{x} \in \mathcal{X}$ the asymptotically stable equilibrium points.

**Theorem 1** *Suppose the perturbed instance $\widetilde{\mathbf{x}}$ is produced by adding perturbation smaller than $\delta$ on the clean instance. If all the clean instances $\mathbf{x} \in \mathcal{X}$ are the asymptotically stable equilibrium points of ODE (1), there exists $\delta > 0$, for each contaminated instance $\hat{\mathbf{x}} \in \{\widetilde{\mathbf{x}} : \widetilde{\mathbf{x}} \in \widetilde{\mathcal{X}}, \widetilde{\mathbf{x}} \notin \mathcal{X}\}$, there exists $\mathbf{x} \in \mathcal{X}$ such that $\lim_{t \to +\infty} ||\mathbf{s}(\hat{\mathbf{x}}, t) - \mathbf{x}|| = 0$.*

See proof in Appendix A.1.

Theorem 1 guarantees that if we make clean instance $\mathbf{x}$ into the asymptotically stable equilibrium point, nonautonomous ODE can shrink the perturbation and make the perturbed instance approach to the clean instance, which could help improve the robustness of the DNN and aid the DNN in defending against adversarial attack. Next, we talk about how to ensure the ODE has asymptotic stability.

## 4 Linearization and Stability of ODE

In order to make the nonautonomous ODE (1) have asymptotic stability, we linearize ODE (1) using Lyapunov's linearization method. We then impose constraints on the linearization to make ODE (1) asymptotically stable.

Suppose $\mathbf{x}^*$ is an equilibrium point of nonautonomous systems (1),

$$\mathbf{h}(\mathbf{x}^*, t) = 0, \forall t \geq 0, \tag{4}$$

where $\mathbf{h}$ is a $C^1$ function. We define

$$\mathbf{A}(t) = \left[ \frac{\partial \mathbf{h}(\mathbf{z}, t)}{\partial \mathbf{z}} \right]_{\mathbf{z} = \mathbf{x}^*}, \tag{5}$$

$$\mathbf{h_r}(\mathbf{z}, t) = \mathbf{h}(\mathbf{z}, t) - \mathbf{A}(t)(\mathbf{z} - \mathbf{x}^*). \tag{6}$$

Then, by the definition of the Jacobian, it follows that for each fixed $t \geq 0$, it is true that

$$\lim_{||\mathbf{z}|| \to \mathbf{x}^*} \frac{||\mathbf{h_r}(\mathbf{z}, t)||}{||\mathbf{z} - \mathbf{x}^*||} = 0. \tag{7}$$

However, it may not be true that

$$\lim_{||\mathbf{z}|| \to \mathbf{x}^*} \sup_{t \geq 0} \frac{||\mathbf{h_r}(\mathbf{z}, t)||}{||\mathbf{z} - \mathbf{x}^*||} = 0. \tag{8}$$

In other words, the convergence in (7) may not be uniform in $t$. Provided that (8) holds, the system

$$\frac{d\mathbf{z}(t)}{dt} = \mathbf{A}(t)(\mathbf{z} - \mathbf{x}^*). \tag{9}$$

is referred to as the linearization of (1) around the equilibrium $\mathbf{x}^*$.

**Theorem 2** *Suppose that (4) holds and $\mathbf{h}(\mathbf{z}, t)$ is continuously differentiable. Define $\mathbf{A}(t)$, $h_r(\mathbf{z}, t)$ as in (5), (6), respectively, and assume that (8) holds and $\mathbf{A}(t)$ is bounded. If $\mathbf{x}^*$ is an exponentially stable equilibrium of the linear system (9), then it is also an exponentially stable equilibrium of the system (1).*

See proof in Appendix A.2.

In order to make all the equilibrium points of (9) asymptotically stable, we deduce the stability of the nonautonomous system (1) by studying only the "frozen" systems; that is, the ODE (1) with time "frozen" at $r$. In other words, if $r \in \mathbb{R}^+$ is any fixed number, we can think of the autonomous system

$$\frac{d\mathbf{z}(t)}{dt} = \mathbf{h}(\mathbf{z}(t), r), \forall t \geq 0, \tag{10}$$

If $\mathbf{x}^*$ is the equilibrium point, then $\mathbf{A} = [\frac{\partial \mathbf{h}(\mathbf{z}, r)}{\partial \mathbf{z}}]_{\mathbf{z}=\mathbf{x}^*}$ is a constant matrix and the linearization (9) becomes $\dot{\mathbf{z}}(t) = \mathbf{A}(\mathbf{z} - \mathbf{x}^*)$. Moreover, we use $\mathbf{s}_r(\mathbf{x}, t)$ to denote the solution of frozen system (10), starting at time 0 and state $\mathbf{x}$, and evaluated at time $t$. Even if each of the frozen systems (10) is exponentially stable, the overall system can be unstable [14]. Next, we will prove that if each frozen system is exponentially stable and the system is slowly varying, the overall system is indeed exponentially stable.

**Theorem 3** *Suppose (i) $\mathbf{h}$ is $C^1$, $\mathbf{h}(\mathbf{x}^*, t) = 0, \forall t \geq 0$, and (ii)* $\sup\limits_{\mathbf{z} \in \mathbb{R}^n} \sup\limits_{t \geq 0} ||\nabla h(\mathbf{z} - \mathbf{x}^*, t)|| = \eta < \infty$,
*(iii) there exist constants $\mu$, $\delta$ such that $|\mathbf{s}_r(\mathbf{z} - \mathbf{x}^*, t)|| \leq \mu ||\mathbf{z} - \mathbf{x}^*|| \exp(-\delta t), \forall t \geq 0, \forall \mathbf{z} \in \mathbb{R}^n, \forall r \in \mathbb{R}^+$. (iv) suppose there is a constant $\epsilon > 0$ such that*

$$||\frac{\partial \mathbf{h}(\mathbf{z} - \mathbf{x}^*, t)}{\partial t}|| \leq \epsilon ||\mathbf{z} - \mathbf{x}^*||, \forall t \geq 0, \forall \mathbf{z} \in \mathbb{R}^n. \tag{11}$$

*Then the nonautonomous systems (1) is exponentially stable provided $\epsilon < \frac{\delta[(p-1)\delta - \eta]}{p\mu^p}$, where $p > 1$ is any number such that $(p-1)\delta - \eta > 0$.*

See proof in Appendix A.3

Based on Theorems 2 and 3, we impose constraints on ODE (1) to be a slowly varying systems and linearize it to be (9). In order to make (9) stable, we transform the study of nonautonomous system (1) to the study of autonomous system (10); in other words, we should make $\dot{\mathbf{z}}(t) = \mathbf{A}(\mathbf{z} - \mathbf{x}^*)$ asymptotically stable. The following two theorems provide ideas for solving this problem.

**Theorem 4 ([14])** *The equation $\dot{\mathbf{z}}(t) = \mathbf{A}\mathbf{z}$ is asymptotically stable if and only if all eigenvalues of $\mathbf{A}$ have negative real parts.*

**Theorem 5 (Levy–Desplanques theorem [15])** *Let $A = [a_{ij}] \in M_n$, if $|a_{ii}| \geq \sum\limits_{i \neq j} |a_{ij}|$ and $a_{ii} \leq 0$ for all $i = 1, \cdots n$, then every eigenvalue of $A$ has a negative real part.*

According to theorem 4, we need to make all eigenvalues of each Jabobian matrix $\mathbf{A}$ of (10) at equilibrium points have negative real parts. In addition, in the light of the Levy–Desplanques theorem [15], rather than computing the eigenvalues, we instead add constraints to the elements of Jacobian matrix $\mathbf{A}$ to make all its eigenvalues have negative real parts. Finally, we make all equilibrium points of the nonautonomous system (1) asymptotically stable by adding constraints on its linearization.

To sum up, we linearize the slowly varying nonautonomous ODE (1) and impose constraints on its linearization (9) to make all clean instances be the asymptotically stable equilibrium points of (1).

## 5 ASODE Architecture

This section proposes a stable neural ODE (ASODE) with asymptotically stable equilibrium points for defending against adversarial attacks. To make all clean instances into the asymptotically stable equilibrium points of neural ODE, we construct an optimization problem according to the constraints mentioned in section 4. To solve this optimization problem, we propose an objective with constraints imposed on its regularizer.

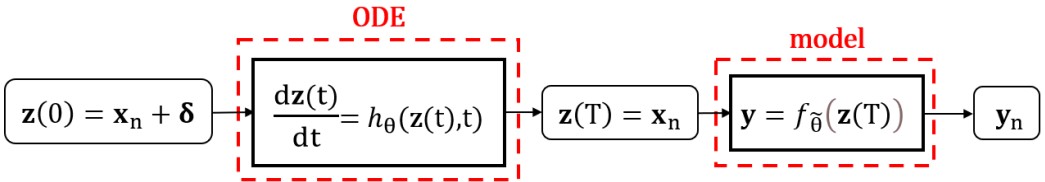

Figure 4: The architecture of our ASODE.

## 5.1 Architecture

Most physical systems are nonlinear and time-varying. If we apply DNNs to dynamic systems, some of them can be described by nonlinear differential equations of the form,

$$\frac{d\mathbf{z}(t)}{dt} = \mathbf{h}_\theta(\mathbf{z}(t), t), \forall t \in [0, T], \tag{12}$$

$$\mathbf{y} = \mathbf{f}_{\widetilde{\theta}}(\mathbf{z}(T)), \mathbf{z}(0) = \mathbf{x}, \mathbf{x} \in \mathcal{X}, \tag{13}$$

where $\mathbf{h}_\theta$ is a neural ODE layer with parameter $\theta$ and defined as $\mathbb{R}^d \times \mathbb{R}^+ \to \mathbb{R}^d$. Similarly, $\mathbf{f}_{\widetilde{\theta}}$ is a neural network with parameter $\widetilde{\theta}$ and defined as $\mathbb{R}^d \to \mathbb{R}^m$. Because the evolutionary time of neural ODE cannot be infinity, we set the final evolutionary time $T$ and $t \in [0, T]$, where $T$ is a positive integer. Inspired by the neural dynamical forms, we propose the ASODE architecture and present it in Figure 4. The ASODE can be simplified as follows: ODE + model. The ODE part can be regarded as a denoiser, which helps to reduce the noise, including the adversarial noise, to rectify the perturbed instance $\mathbf{x}_n + \delta$. The output of ODE is a rectified instance $\mathbf{z}(T) = \mathbf{x}_n$ when $T$ is sufficiently large. Subsequently, the rectified instance $\mathbf{z}(T)$ enters the grafted model $\mathbf{f}_{\widetilde{\theta}}$, and the final output label is not affected.

## 5.2 Asymptotic Stable Guarantee

As mentioned in section 3, we make all $\mathbf{x} \in \mathcal{X}$ the asymptotically stable equilibrium points of neural ODE (12). According to the analysis in section 4, we transform the slowly time-varying ODE (12) to the linear time-varying state equation $\dot{\mathbf{z}}(t) = \mathbf{A}(t)(\mathbf{z}(t) - \mathbf{x})$ and simplify the nonautonomous system (12) to an autonomous system $\frac{d\mathbf{z}(t)}{dt} = \mathbf{h}_\theta(\mathbf{z}(t), r)$. We then make $\dot{\mathbf{z}}(t) = \mathbf{A}(\mathbf{z} - \mathbf{x})$ asymptotically stable, where $\mathbf{A} = [\frac{\partial \mathbf{h}_\theta(\mathbf{z}, r)}{\partial \mathbf{z}}]_{\mathbf{z}=\mathbf{x}}$. Next, we apply the Levy–Desplanques theorem to add constraints on the elements of Jacobian matrix $\mathbf{A}$ to make all its eigenvalues have negative real parts. Therefore, we have the following optimization problem for the ODE part of ASODE:

$$\min_{\theta} \mathbb{E}_{\mathbf{x} \sim \mathcal{D}_{\mathbf{x}}} \mathbb{E}_{\widetilde{\mathbf{x}} \sim \mathcal{D}_{\widetilde{\mathbf{x}}}} \left[ I_{B_\delta(\mathbf{x})}(\widetilde{\mathbf{x}}) \Phi(\mathbf{z}(T), \mathbf{x}) \right] \tag{14}$$

$$s.t. \mathbb{E}[\mathbf{h}_\theta(\mathbf{x}, t)] \le \epsilon_1,, \tag{15}$$

$$\mathbb{E}\left[ ||\frac{\partial \mathbf{h}_\theta(\mathbf{z}, t)}{\partial t}|| / ||\mathbf{z}|| \right] \le \epsilon_2 \tag{16}$$

$$\mathbb{E}[\nabla \mathbf{h}_\theta(\mathbf{x}, t)]_{ii} < 0, \tag{17}$$

$$\mathbb{E}\left[ |[\nabla \mathbf{h}_\theta(\mathbf{x}, t)]_{ii}| - \sum_{j \neq i} |[\nabla \mathbf{h}_\theta(\mathbf{x}, t)]_{ij}| \right] > 0, \tag{18}$$

$$\forall t \in [0, T] \quad and \quad i, j = 1, 2, \cdots, d,$$

where $\Phi$ is the cross-entropy loss, $\epsilon_1$ and $\epsilon_2$ are small constants, and $I_{B_\delta(\mathbf{x})}(\widetilde{\mathbf{x}}) = \begin{cases} 1, \widetilde{\mathbf{x}} \in B_\delta(\mathbf{x}), \\ 0, \widetilde{\mathbf{x}} \notin B_\delta(\mathbf{x}). \end{cases}$

The optimization objective (14) can be illustrated as follows: For any $\mathbf{x} \in \mathcal{X}$, if the perturbed instance $\widetilde{\mathbf{x}} \in B_\delta(\mathbf{x})$, we want the the neural ODE (12) to make $\widetilde{\mathbf{x}}$ converge to $\mathbf{x}$. Therefore, our objective is to minimize the distance between $\mathbf{s}(\widetilde{\mathbf{x}}, T)$ and $\mathbf{x}$. The constraint (15) means that we set each $\mathbf{x} \in \mathcal{X}$ as the asymptotically stable equilibrium point of neural ODE (12), which requires $\mathbf{h}_\theta(x, t) = 0$ according to Definition 1. The constraint (16) implies that we make the neural ODE

(12) the slowly time varying system, which requires $||\frac{\partial \mathbf{h}_\theta(\mathbf{z},t)}{\partial t}|| \leq \epsilon||\mathbf{z}||$ according to the formulation (11) in Theorem 3. The other conditions for Theorem 3 are implicitly satisfied when $t$ is bounded. Besides, the constraints (17) and (18) follows the Levy–Desplanques Theorem 5 to make neural ODE (12) asymptotically stable.

## 5.3 Implementation

Rather than solving the optimization problem (14), we optimize the following empirical Lagrangian (24) with a training set $(\mathbf{x}_n, \mathbf{y}_n)$, $n = 1, 2, \cdots, N$. Moreover, $\mathbf{x}_n^{(1)}, \mathbf{x}_n^{(2)}, \cdots, \mathbf{x}_n^{(K)}$ are $K$ samples from $B_\delta(\mathbf{x}_n)$ used to represent $\widetilde{\mathbf{x}}$. In addition, $\tau = 0, 1, \cdots, T-1$ are the discrete time points. The correspondence and transformation between (14) and (24) are following:

$$\frac{1}{N} \sum_{n=1}^{N} \Phi(\mathbf{s}(\mathbf{x}_n, T), \mathbf{x}_n) + \frac{1}{NK} \sum_{n=1}^{N} \sum_{k=1}^{K} \Phi(\mathbf{s}(\mathbf{x}_n^{(k)}, T), \mathbf{x}_n), \tag{19}$$

$$\frac{1}{NT} \sum_{n=1}^{N} \sum_{\tau=0}^{T-1} ||\mathbf{h}_\theta(\mathbf{x}_n, \tau)||, \tag{20}$$

$$\frac{1}{NT} \sum_{n=1}^{N} \sum_{\tau=0}^{T-1} ||(\frac{\partial \mathbf{h}(\mathbf{s}(\mathbf{x}_n, t), t)}{\partial t})_{t=\tau}|| \Big/ ||\mathbf{s}(\mathbf{x}_n, \tau)||, \tag{21}$$

$$\frac{1}{NT} \sum_{n=1}^{N} \sum_{\tau=0}^{T-1} \sum_{i=1}^{d} -|\nabla \mathbf{h}_\theta(\mathbf{x}_n, \tau)]_{ii}|, \tag{22}$$

$$\frac{1}{NT} \sum_{n=1}^{N} \sum_{\tau=0}^{T-1} \sum_{i=1}^{d} \Big( -|\nabla \mathbf{h}_\theta(\mathbf{x}_n, \tau)]_{ii}| + \sum_{j=1, j \neq i}^{d} |\nabla \mathbf{h}_\theta(\mathbf{x}_n, \tau)]_{ij}| \Big). \tag{23}$$

The constraints (15), (16), (17) and (18) correspond to the discretization forms (20), (21), (22) and (23) respectively. Furthermore, the objective (14) corresponds to (19). Based on the analysis of the optimization problem (14), we get that the smaller the values (19)- (23) are, the better the neural ODE is. Therefore, we construct the following empirical loss $L_{ODE}$:

$$L_{ODE} = \min_{\theta} \frac{1}{N} \sum_{n=1}^{N} \Bigg\{ \Phi(\mathbf{s}(\mathbf{x}_n, T), \mathbf{x}_n) + \frac{1}{K} \sum_{k=1}^{K} \Phi(\mathbf{s}(\mathbf{x}_n^{(k)}, T), \mathbf{x}_n) + \frac{1}{T} \sum_{\tau=0}^{T-1} \Bigg[ \tag{24}$$

$$\alpha_1 \Bigg( ||\mathbf{h}_\theta(\mathbf{x}_n, \tau)|| + ||(\frac{\partial \mathbf{h}(\mathbf{s}(\mathbf{x}_n, t), t)}{\partial t})_{t=\tau}|| \Big/ ||\mathbf{s}(\mathbf{x}_n, \tau)|| \Bigg) +$$

$$\alpha_2 \Bigg( \exp \big( \sum_{i=1}^{d} -|\nabla \mathbf{h}_\theta(\mathbf{x}_n, \tau)]_{ii}| \big) + \exp \big( \sum_{i=1}^{d} (-|\nabla \mathbf{h}_\theta(\mathbf{x}_n, \tau)]_{ii}| + \sum_{j=1, j \neq i}^{d} |\nabla \mathbf{h}_\theta(\mathbf{x}_n, \tau)]_{ij}|) \big) \Bigg) \Bigg] \Bigg\}.$$

According to our proposed ASODE, finding the optimal $\theta$ is the first phase in making the ODE (12) a good denoiser that is capable of reducing the adversarial noise. We then fix the neural ODE (12) and train the grafted model as: $\min_{\widetilde{\theta}} \mathbb{E}_{(\mathbf{x}, \mathbf{y}) \sim \mathcal{D}} [\Phi(\mathbf{f}_{\widetilde{\theta}}(\mathbf{z}(T)), \mathbf{y})]$. Its empirical loss is $L_{model}$:

$$L_{model} = \min_{\widetilde{\theta}} \frac{1}{N} \sum_{n=1}^{N} \Phi(\mathbf{f}_{\widetilde{\theta}}(\mathbf{s}(\mathbf{x}_n, T)), \mathbf{y}_n). \tag{25}$$

In sum, our ASODE has two parts: the neural part plays the role of denoiser, and the grafted model has the ability of recognition. In order to obtain the dynamic systems as we expected, we propose two related optimization problems (24) and (25). Finally, we solve these two optimization problems (24) and (25) to make all the clean instances be the asymptotically stable equilibrium points of the neural ODE (12), thereby improving the robustness of our ASODE. The pseudo code of ASODE algorithm is illustrated in Appendix B, which is from bringing the process above together.

# 6 Experiments

In this section, we conduct experiments on the CIFAR-10 [16] and MNIST [17] datasets to evaluate the robustness of ASODE under different adversarial attacks. We follow the standard training, validation, and test splits in our experiments. Moreover, we compare the robustness of ASODE with ODE-Net [12], TisODE-Net [10], and SODEF [11].

## 6.1 Setup

We apply different attack methods to attack our model, namely FGSM [29] and PGD [28]. Following the same experimental settings as for SODEF, we implement the following settings. 1. For the CIFAR-10 task, we use the model provided in [36]. 2. For the MNIST task, we use the ResNet18 model provided in PyTorch. The baselines are re-implemented according to the original paper. The neural ODE function $\mathbf{h}_\theta$ is made up of three fully connected layers whose input and output layers have the same dimension. During the training of ASODE, we first train the neural ODE for 50 epochs, after which we fix $\mathbf{h}_\theta$ and train $\mathbf{f}_{\widetilde{\theta}}$ for another 100 epochs. We set the parameters $\alpha_1 = 0.1$ and $\alpha_2 = 0.05$ when training ASODE. In the below, the best results are marked in **bold**.

## 6.2 Performance Under PGD and FGSM Attacks

We evaluate the robustness of our ASODE under PGD attack and FGSM attack. Specifically, we compare the robustness of ASODE with ODE-Net, TisODE-Net and SODEF on the MNIST and CIFAR-10. To facilitate fair comparison, we set $T = 5$ based on the original papers of the comparison models. The results on MNIST and CIFAR-10 are presented in Table 1 and Table 2 respectively. As the results show, under PGD and FGSM attacks, our model ASODE outperforms the state-of-the-art SODEF; moreover, ASODE also performs much better than ODE-Net and TisODE-Net. For example, compared with SODEF under PGD attack on the MNIST and CIFAR-10, our ASODE improves the classification accuracy of adversarial examples by $1.60\%$ and $1.76\%$ respectively. Furthermore, compared with TisODE-Net on the same datasets, the classification accuracy obtained by SODEF are increased by $45.07\%$ and $53.53\%$ respectively. The "NO ODE" in Tables 1 and 2 corresponds to ASODE without the ODE part. For ablation studies, we find that ASODE is more robust than NO ODE, which indicates that ODE part (12) helps improve the robustness.

Table 1: Classification accuracy (%) on adversarial MNIST examples with $\mathcal{L}_\infty$ norm, $\epsilon = 0.3$.

| Attack | No ODE | ODE | TisODE | SODEF | ASODE |
|--------|--------|-------|--------|-------|--------|
| None | **99.45** | 99.40 | 99.41 | 99.44 | 99.44 |
| FGSM | 9.87 | 29.51 | 36.82 | 63.36 | **65.13** |
| PGD | 0.36 | 1.68 | 1.78 | 45.25 | **46.85** |

Table 2: Classification accuracy (%) on adversarial CIFAR-10 examples with $\mathcal{L}_\infty$ norm, $\epsilon = 0.031$.

| Attack | No ODE | ODE | TisODE | SODEF | ASODE |
|--------|--------|-------|--------|-------|--------|
| None | **95.20** | 94.76 | 95.12 | 95.00 | 95.16 |
| FGSM | 47.54 | 45.12 | 43.37 | 68.05 | **69.94** |
| PGD | 3.18 | 3.50 | 3.82 | 55.59 | **57.35** |

# 7 Conclusion

In this paper, we develop a novel nonautonomous time-slowing varying neural ODE, ASODE, that makes all clean instances be its asymptotically stable equilibrium points. ASODE shrinks the adversarial noise and forces the adversarial instance to be close to the clean instance. The experimental results show that ASODE improves the robustness against adversarial attacks and outperforms the state-of-the-art methods.

# Acknowledgements

This work is supported by the National Natural Science Foundation of China under Grant 61976161.

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
