# Defending Against Adversarial Attacks via Neural Dynamic System (Appendix)

## A Proof of Proposition and Theorem

$$\frac{d\mathbf{z}(t)}{dt} = \mathbf{h}(\mathbf{z}(t), t). \tag{1}$$

Assume $\mathbf{x}^*$ is an equilibrium of (1). We have the same meaning for $\mathbf{x}^*$ in our Appendix.

### A.1 Proof of Theorem 1

**Theorem 1** *Suppose that the perturbed instance $\widetilde{\mathbf{x}}$ is produced by adding perturbations smaller than $\delta$ on a clean instance. If all the clean instances $\mathbf{x} \in \mathcal{X}$ are the asymptotically stable equilibrium points of ODE (1), there exists $\delta > 0$, for each contaminated instance $\hat{\mathbf{x}} \in \{\widetilde{\mathbf{x}} : \widetilde{\mathbf{x}} \in \widetilde{\mathcal{X}}, \widetilde{\mathbf{x}} \notin \mathcal{X}\}$, there exists $\mathbf{x} \in \mathcal{X}$ such that $\lim_{t \to +\infty} ||\mathbf{s}(\hat{\mathbf{x}}, t) - \mathbf{x}|| = 0$.*

**Proof**:

According to the definition of asymptotic stability, A constant vector of (1) is asymptotically stable if it is stable and attractive. Based on the definition of stability of (1), for each $\epsilon > 0$ and each $t_0 \in \mathbb{R}^+$, there exists $\delta_1 = \delta(\epsilon, 0)$ such that

$$\forall \widetilde{\mathbf{x}} \in B_{\delta_1}(\mathbf{x}) \Rightarrow ||\mathbf{s}(\widetilde{\mathbf{x}}, t) - \mathbf{x}|| < \epsilon, \forall t \geq t_0.$$

Based on the Attractivity Definition (1), there exists $\delta_2 = \delta(0) > 0$ such that

$$\widetilde{\mathbf{x}} \in B_{\delta_2}(\mathbf{x}), \lim_{t \to +\infty} ||\mathbf{s}(\widetilde{\mathbf{x}}; t) - \mathbf{x}|| = 0.$$

We make $\delta = \min\{\delta_1, \delta_2\}$. Because the perturbed instance $\widetilde{\mathbf{x}}$ is produced by adding perturbation smaller than $\delta$ on the clean instance, then for each contaminated instance $\hat{\mathbf{x}} \in \{\widetilde{\mathbf{x}} : \widetilde{\mathbf{x}} \in \widetilde{\mathcal{X}}, \widetilde{\mathbf{x}} \notin \mathcal{X}\}$, there exists clean instance $\mathbf{x} \in \mathcal{X}$ such that $\hat{\mathbf{x}} \in B_\delta(\mathbf{x})$. Because the clean instance $\mathbf{x}$ is an asymptotically stable equilibrium point of (1), we have

$$\lim_{t \to +\infty} ||\mathbf{s}(\hat{\mathbf{x}}, t) - \mathbf{x}|| = 0.$$

∎

### A.2 Proof of Theorem 2

suppose $\mathbf{x}^*$ is an equilibrium point of nonautonomous systems (1),

$$\mathbf{h}(\mathbf{x}^*, t) = 0, \forall t \geq 0, \tag{2}$$

and $\mathbf{h}$ is a $C^1$ function. Define

$$\mathbf{A}(t) = \left[\frac{\partial \mathbf{h}(\mathbf{z}, t)}{\partial \mathbf{z}}\right]_{\mathbf{z}=\mathbf{x}^*}, \tag{3}$$

$$\mathbf{h_r}(\mathbf{z}, t) = \mathbf{h}(\mathbf{z}, t) - \mathbf{A}(t)(\mathbf{z} - \mathbf{x}^*). \tag{4}$$

Then, by the definition of the Jacobian, it follows that for each fixed $t \geq 0$, it is true that

$$\lim_{||\mathbf{z}|| \to \mathbf{x}^*} \frac{||\mathbf{h_r}(\mathbf{z}, t)||}{||\mathbf{z} - \mathbf{x}^*||} = 0. \tag{5}$$

24 However, it may not be true that

$$\lim_{||\mathbf{z}||\to\mathbf{x}^*}\sup_{t\geq 0}\frac{||\mathbf{h_r}(\mathbf{z},t)||}{||\mathbf{z}-\mathbf{x}^*||}=0. \tag{6}$$

25 In other words, the convergence in (5) may not be uniform in $t$. Provided (6) holds, the system will

$$\frac{d\mathbf{z}(t)}{dt}=\mathbf{A}(t)(\mathbf{z}-\mathbf{x}^*). \tag{7}$$

26 is called the linearization of (1) around the equilibrium $\mathbf{x}^*$.

27 **Lemma 1 ([1])** *Suppose $Q:\mathbb{R}^+\to\mathbb{R}^{d\times d}$ is continuous and bounded, and that the equilibrium $\mathbf{x}^*$*
28 *of (7) is uniformly asymptotically stable. Then, for each $t\geq 0$, the matrix is as follows:*

$$\mathbf{P}(t)=\int_t^{+\infty}\Phi^\top(\tau,t)Q(\tau)\Phi(\tau,t)d\tau$$

29 *is well defined and $\mathbf{P}(t)$ is bounded as a function of $t$. Here, $\Phi(\cdot,\cdot)$ is the state transition matrix of*
30 *system (7) defined in [1].*

31 **Lemma 2 ([2])** *Suppose that $Q:\mathbb{R}^+\to\mathbb{R}^{d\times d}$ is continuous and bounded and that the equilibrium*
32 *$\mathbf{x}^*$ of (7) is uniformly asymptotically stable. Moreover, if the following conditions also hold:*

33 *(i) $\mathbf{Q}(t)$ is symmetric and positive definite for each $t\geq 0$ and there exists a constant $\alpha>0$ such that*

$$\alpha(\mathbf{z}-\mathbf{x}^*)^\top(\mathbf{z}-\mathbf{x}^*)\leq(\mathbf{z}-\mathbf{x}^*)^\top\mathbf{Q}(t)(\mathbf{z}-\mathbf{x}^*),\forall\mathbf{z}\in\mathbb{R}^d,\forall t\geq 0.$$

34 *(ii) The matrix $\mathbf{A}(t)$ in (7) is bounded; i,e,*

$$m_0:=\sup_{t\geq 0}||\mathbf{A}(t)||<+\infty,$$

35 *under these conditions, the matrix $\mathbf{P}(t)$ defined in Lemma 1 is positive definite for each $t\geq 0$;*
36 *moreover, there exists a constant $\beta>0$ such that*

$$\beta(\mathbf{z}-\mathbf{x}^*)^\top(\mathbf{z}-\mathbf{x}^*)\leq(\mathbf{z}-\mathbf{x}^*)^\top\mathbf{P}(t)(\mathbf{z}-\mathbf{x}^*),\forall\mathbf{z}\in\mathbb{R}^d,\forall t\geq 0.$$

37 **Lemma 3 ([3])** *Suppose there exist constants $a,b,c,r>0$, $p\geq 1$, and a $C^1$ function $V:\mathbb{R}^d\times$*
38 *$\mathbb{R}^+\to\mathbb{R}$ such that*

$$a||\mathbf{z}-\mathbf{x}^*||^p\leq V(\mathbf{z}-\mathbf{x}^*,t)\leq b||\mathbf{z}-\mathbf{x}^*||^p,\mathbf{z}\in\forall\mathbf{B}_r(\mathbf{x}^*),\forall t\geq 0,$$
$$\dot{V}(\mathbf{z}-\mathbf{x}^*,t)\leq -c||\mathbf{z}-\mathbf{x}^*||^p,\forall\mathbf{z}\in\mathbf{B}_r(\mathbf{x}^*),\forall t\geq 0.$$

39 *Then the equilibrium $\mathbf{x}^*$ is exponentially stable.*

40 **Theorem 2** *Suppose that (2) holds and $\mathbf{h}(\mathbf{z},t)$ is continuously differentiable. Define $\mathbf{A}(t)$, $h_r(\mathbf{z},t)$*
41 *as in (3), (4), respectively, and assume that (6) holds and $\mathbf{A}(t)$ is bounded. If $\mathbf{x}^*$ is an exponentially*
42 *stable equilibrium of the linear system (7), then it is also an exponentially stable equilibrium of the*
43 *system (1).*

44 **Proof**: Since $\mathbf{A}(t)$ is bounded and the equilibrium $\mathbf{x}^*$ is uniformly asymptotically stable, from
45 Lemma 2, that the matrix

$$\mathbf{P}(t)=\int_t^{+\infty}\Phi^\top(\tau,t)\Phi(\tau,t)d\tau \tag{8}$$

46 is well-defined for $t\geq 0$; moreover, there exist constants $\alpha,\beta>0$ such that

$$\alpha(\mathbf{z}-\mathbf{x}^*)^\top(\mathbf{z}-\mathbf{x}^*)\leq(\mathbf{z}-\mathbf{x}^*)^\top\mathbf{P}(t)(\mathbf{z}-\mathbf{x}^*)\leq\beta(\mathbf{z}-\mathbf{x}^*)^\top(\mathbf{z}-\mathbf{x}^*),\forall\mathbf{z}\in\mathbb{R}^d,\forall t\geq 0. \tag{9}$$

47 Hence the function

$$V(\mathbf{z} - \mathbf{x}^*, t) = (\mathbf{z} - \mathbf{x}^*)^\top \mathbf{P}(t)(\mathbf{z} - \mathbf{x}^*)$$

48 is a decrescent positive definite function. Calculating $\dot{V}$ for the system (1) gives

$$
\begin{aligned}
\dot{V}(\mathbf{z} - \mathbf{x}^*, t) &= (\mathbf{z} - \mathbf{x}^*)^\top \dot{\mathbf{P}}(t)(\mathbf{z} - \mathbf{x}^*) + \mathbf{h}(\mathbf{z} - \mathbf{x}^*, t)\mathbf{P}(t)(\mathbf{z} - \mathbf{x}^*) \\
&\quad + (\mathbf{z} - \mathbf{x}^*)^\top \mathbf{P}(t)\mathbf{h}((\mathbf{z} - \mathbf{x}^*), t) \\
&= (\mathbf{z} - \mathbf{x}^*)^\top [\dot{\mathbf{P}}(t) + \mathbf{A}^\top(t)\mathbf{P}(t) + \mathbf{P}(t)\mathbf{A}(t)](\mathbf{z} - \mathbf{x}^*) \\
&\quad + 2(\mathbf{z} - \mathbf{x}^*)^\top \mathbf{P}(t)\frac{\partial \mathbf{h}(\mathbf{z} - \mathbf{x}^*, t)}{\partial t}.
\end{aligned}
$$

49 However, from (8) it can be easily shown that

$$\dot{\mathbf{P}}(t) + \mathbf{A}^\top(t)\mathbf{P}(t) + \mathbf{P}(t)\mathbf{A}(t) = -\mathbf{I}.$$

50 where $\mathbf{I}$ is the identity matrix. Therefore,

$$\dot{V}(\mathbf{z} - \mathbf{x}^*, t) = -(\mathbf{z} - \mathbf{x}^*)^\top(\mathbf{z} - \mathbf{x}^*) + 2(\mathbf{z} - \mathbf{x}^*)^\top \dot{\mathbf{P}}(t)\frac{\partial \mathbf{h}(\mathbf{z} - \mathbf{x}^*, t)}{\partial t}.$$

51 In the view of (6), one can pick a number $r > 0$ and a $\rho < 0.5$ such that

$$||\frac{\partial \mathbf{h}(\mathbf{z} - \mathbf{x}^*, t)}{\partial t}|| \leq \frac{\rho}{\beta}||\mathbf{z} - \mathbf{x}^*||, \forall \mathbf{z} \in \mathbf{B}_r(\mathbf{x}^*), \forall t \geq 0. \tag{10}$$

52 Then (10) and (9) together imply that

$$|2(\mathbf{z} - \mathbf{x}^*)^\top \mathbf{P}(t)\frac{\partial \mathbf{h}(\mathbf{z} - \mathbf{x}^*, t)}{\partial t}| \leq \frac{2\rho}{\beta}(\mathbf{z} - \mathbf{x}^*)^\top(\mathbf{z} - \mathbf{x}^*), \forall \mathbf{z} \in \mathbf{B}_r(\mathbf{x}^*), \forall t \geq 0.$$

53 therefore,

$$\dot{V}(\mathbf{z} - \mathbf{x}^*, t) \leq -(1 - 2\rho)(\mathbf{z} - \mathbf{x}^*)^\top(\mathbf{z} - \mathbf{x}^*), \mathbf{z} \in \mathbf{B}_r(\mathbf{x}^*), \forall t \geq 0.$$

54 this shows that $-\dot{V}$ is an locally positive definite function. Based on Lemma 3, we conclude that $\mathbf{x}^*$
55 is an exponentially stable equilibrium.

56 ∎

## A.3 Proof of Theorem 3

58 **Lemma 4 (Gronwall [4])** *Suppose $a(t)$: $\mathbb{R}^+ \to \mathbb{R}^+$ is a continuous function and $b, c \geq 0$ are given*
59 *constants. Under these conditions, if*

$$a(t) \leq b + \int_0^t ca(\tau)d\tau, \forall t \geq 0,$$

60 *then*

$$a(t) \leq b \exp(ct), \forall t \geq 0.$$

61 **Lemma 5 ([2])** *Consider the system (1), and suppose $\mathbf{h}$ is $C^k$, and that $\mathbf{h}(\mathbf{x}^*, t) = 0, \forall t \geq 0$.*
62 *Suppose that there exist constants $\mu, \delta, r > 0$ such that*

$$||\mathbf{s}(\mathbf{z} - \mathbf{x}^*, t, \tau)|| \leq \mu||\mathbf{z} - \mathbf{x}^*|| \exp(-\delta(\tau - t)), \forall \tau \geq t \geq 0, \mathbf{z} \in \mathbf{B}_r(\mathbf{x}^*).$$

63 *Finally, suppose that, for some finite constant $\eta$,*

$$||\nabla \mathbf{h}(\mathbf{z} - \mathbf{x}^*, t)|| \leq \eta, \forall t \geq 0, \mathbf{z} \in \mathbf{B}_{\mu r}(\mathbf{x}^*)$$

64 *Under these conditions, there exist a $C^k$ function $V : \mathbb{R}^d \times \mathbb{R}^+ \to \mathbb{R}$ and constants $a, b, c, m >$*
65 *$0, p > 1$, such that*

66 $$a||\mathbf{z} - \mathbf{x}^*||^p \leq V(\mathbf{z} - \mathbf{x}^*, t) \leq b||\mathbf{z} - \mathbf{x}^*||^p, \dot{V}(\mathbf{z} - \mathbf{x}^*, t) \leq -c||\mathbf{z} - \mathbf{x}^*||^p, \forall \mathbf{z} \in \mathbf{B}_r(\mathbf{x}^*), \forall t \geq 0,$$

$$||\frac{\partial V(\mathbf{z} - \mathbf{x}^*, t)}{\partial \mathbf{z}}|| \leq m||\mathbf{z} - \mathbf{x}^*||^{p-1}, \forall \mathbf{z} \in \mathbf{B}_r(\mathbf{x}^*), \forall t \geq 0.$$

We first prove the general case of the Theorem 3 in our main paper. We introduce the frozen system.

$$\frac{d\mathbf{z}(t)}{dt} = \mathbf{h}(\mathbf{z}(t), r). \tag{11}$$

we use $\mathbf{s}_r(\mathbf{z}, \tau, t)$ to denote the frozen system (11) solution, starting at time $\tau$ and state $\mathbf{z}$, and evaluated at time $t$.

**Theorem 3 (general)** *Consider the system (1). Suppose (i) $\mathbf{h}$ is $C^1$ and (ii)*

$$\sup_{\mathbf{z} \in \mathbb{R}^n} \sup_{t \geq 0} ||\nabla \mathbf{h}(\mathbf{z} - \mathbf{x}^*, t)|| = \eta < \infty. \tag{12}$$

*(iii) there exist constants $\mu$, $\delta$ such that*

$$||\mathbf{s}_r(\mathbf{z} - \mathbf{x}^*, \tau, t)|| \leq \mu ||\mathbf{z} - \mathbf{x}^*|| \exp\left(-\delta(t - \tau)\right), \forall t \geq \tau \geq 0, \forall \mathbf{z} \in \mathbb{R}^n, r \in \mathbb{R}^+. \tag{13}$$

*(iv), suppose that there is a constant $\epsilon > 0$ such that*

$$||\frac{\partial \mathbf{h}(\mathbf{z} - \mathbf{x}^*, t)}{\partial t}|| \leq \epsilon ||\mathbf{z} - \mathbf{x}^*||, \forall t \geq 0, \forall \mathbf{z} \in \mathbb{R}^n. \tag{14}$$

*Then the nonautonomous system (1) is exponentially stable, provided that*

$$\epsilon < \frac{\delta[(p-1)\delta - \eta]}{p\mu^p}, \tag{15}$$

*where $p > 1$ is any number such that $(p-1)\delta - \eta > 0$.*

**Proof**:

We begin by estimating the rate of variation of the function $\mathbf{s}_r(\mathbf{z} - \mathbf{x}^*, 0, t)$ with respect to $r$. From (11), it follows that

$$\mathbf{s}_r(\mathbf{z} - \mathbf{x}^*, 0, t) = \mathbf{z} - \mathbf{x}^* + \int_0^t \mathbf{h}(\mathbf{s}_r(\mathbf{z} - \mathbf{x}^*, 0, \sigma), r)d\sigma.$$

Differentiating with respect $r$ gives

$$\frac{\partial \mathbf{s}_r(\mathbf{z} - \mathbf{x}^*, 0, t)}{\partial r} = \int_0^t (\frac{\partial \mathbf{h}(\mathbf{s}_r(\mathbf{z} - \mathbf{x}^*, 0, \sigma), r)}{\partial r} + \frac{\partial \mathbf{h}(\mathbf{s}_r(\mathbf{z} - \mathbf{x}^*, 0, \sigma), r)}{\partial \mathbf{s}_r}\frac{\partial \mathbf{s}_r(\mathbf{z} - \mathbf{x}^*, 0, \sigma)}{\partial r})d\sigma. \tag{16}$$

For conciseness, define

$$g(t) = ||\frac{\partial \mathbf{s}_r(\mathbf{z} - \mathbf{x}^*, 0, t)}{\partial r}||,$$

and note from (14) that

$$||\frac{\partial \mathbf{h}(\mathbf{s}_r(\mathbf{z} - \mathbf{x}^*, 0, \sigma), r)}{\partial t}|| \leq \epsilon ||\mathbf{s}_r(\mathbf{z} - \mathbf{x}^*, 0, \sigma)|| \leq \epsilon \mu ||\mathbf{z} - \mathbf{x}^*|| \exp\left(-\delta\sigma\right). \tag{17}$$

Using (12),(17) in (16), we have

$$g(t) \leq \int_0^t \epsilon \mu ||\mathbf{z} - \mathbf{x}^*|| \exp\left(-\delta\sigma\right)d\sigma + \int_0^t \eta g(\sigma)d\sigma \tag{18}$$

$$\leq \frac{\epsilon \mu ||\mathbf{z} - \mathbf{x}^*||}{\delta} + \int_0^t \eta g(\sigma)d\sigma.$$

Applying Lemma 4 to (18) gives

$$||\frac{\partial \mathbf{s}_r(\mathbf{z} - \mathbf{x}^*, 0, t)}{\partial r}|| = g(t) \leq \frac{\epsilon \mu ||\mathbf{z} - \mathbf{x}^*||}{\delta} \exp\left(\eta t\right), \forall t \geq 0. \tag{19}$$

For each $r \geq 0$, define a Lyapunov function $V_r : \mathbb{R}^d \to \mathbb{R}$ for the system (11). Select $p > 1 + \frac{\eta}{\delta}$, and define

$$V_r(\mathbf{z}) = \int_0^{+\infty} ||\mathbf{s}_r(\mathbf{z} - \mathbf{x}^*, 0, t)||^p dt.$$

Since the system (11) is autonomous. we replace $r$ by $\tau$, and define $V : \mathbb{R}^d \times \mathbb{R}^+ \to \mathbb{R}$ by

$$V(\mathbf{z}, \tau) = \int_0^{+\infty} ||\mathbf{s}_\tau(\mathbf{z} - \mathbf{x}^*, 0, t)||^p dt, \tag{20}$$

then, as shown in the lemma 5.

$$\frac{1}{2^{(p+1)}\eta\mu}||\mathbf{z} - \mathbf{x}^*||^p \leq V(\mathbf{z} - \mathbf{x}^*, \tau) \leq \frac{\mu^p}{p\delta}||\mathbf{z} - \mathbf{x}^*||^p. \tag{21}$$

$$\frac{\partial V(\mathbf{z} - \mathbf{x}^*, \tau)}{\partial \mathbf{z}}\mathbf{h}(\mathbf{z} - \mathbf{x}^*, \tau) = -||\mathbf{z} - \mathbf{x}^*||^p.$$

Let us compute the derivative $\dot{V}(\mathbf{z} - \mathbf{x}^*, \tau)$ along the trajectories of (1). By definition

$$\dot{V}(\mathbf{z} - \mathbf{x}^*, \tau) = \frac{\partial V(\mathbf{z} - \mathbf{x}^*, \tau)}{\partial \mathbf{z}}\mathbf{h}(\mathbf{z} - \mathbf{x}^*, \tau) + \frac{\partial V(\mathbf{z} - \mathbf{x}^*, \tau)}{\partial \tau} = \frac{\partial V(\mathbf{z} - \mathbf{x}^*, \tau)}{\partial \tau} - ||\mathbf{z} - \mathbf{x}^*||^p. \tag{22}$$

It only remains to estimate $\frac{\partial V(\mathbf{z}, \tau)}{\partial \tau}$, let $\gamma := \frac{p}{2}$, then, from (20),

$$\frac{\partial V(\mathbf{z} - \mathbf{x}^*, \tau)}{\partial \tau} = \int_0^{+\infty} \frac{\partial [\mathbf{s}_\tau^\top(\mathbf{z} - \mathbf{x}^*, 0, t)\mathbf{s}_\tau(\mathbf{z} - \mathbf{x}^*, 0, t)]^\gamma}{\partial \tau} dt$$

$$= \int_0^{+\infty} 2\gamma[\mathbf{s}_\tau^\top(\mathbf{z} - \mathbf{x}^*, 0, t)\mathbf{s}_\tau(\mathbf{z} - \mathbf{x}^*, 0, t)]^{\gamma-1}\mathbf{s}_\tau^\top(\mathbf{z} - \mathbf{x}^*, 0, t)\frac{\partial \mathbf{s}_\tau(\mathbf{z}, 0, t)}{\partial \tau} dt$$

$$|\frac{\partial V(\mathbf{z} - \mathbf{x}^*, \tau)}{\partial \tau}| \leq \int_0^{+\infty} 2\gamma||\mathbf{s}_\tau(\mathbf{z} - \mathbf{x}^*, 0, t)||^{\gamma-1}||\frac{\partial \mathbf{s}_\tau(\mathbf{z} - \mathbf{x}^*, 0, t)}{\partial \tau}||dt.$$

Now use the bound in (13) for $||\mathbf{s}_\tau(\mathbf{z} - \mathbf{x}^*, 0, t)||$ and (19) for $\frac{\partial \mathbf{s}_\tau(\mathbf{z} - \mathbf{x}^*, 0, t)}{\partial \tau}$, and note that $2\gamma = p$. This gives

$$|\frac{\partial V(\mathbf{z} - \mathbf{x}^*, \tau)}{\partial \tau}| \leq \int_0^{+\infty} p\mu^{p-1}||\mathbf{z} - \mathbf{x}^*||^{p-1}\frac{\epsilon\mu||\mathbf{z} - \mathbf{x}^*||}{\delta} \exp\left[-(p-1)\delta t + \eta t\right]dt$$

$$= \frac{p\epsilon\mu^p}{\delta[(p-1)\delta - \eta]}||\mathbf{z} - \mathbf{x}^*||^p.$$

Let $m$ denote the constant multiplying $||\mathbf{z} - \mathbf{x}^*||^p$ on the right side, and note that $m < 1$ by (15). Finally, from (22)

$$\dot{V}(\mathbf{z} - \mathbf{x}^*, t) \leq -(1 - m)||\mathbf{z} - \mathbf{x}^*||^p. \tag{23}$$

Now (21) and (23) show that $V$ is a suitable Lyapunov function for applying the Lemma 5 to conclude the exponential stability. And we get Theorem 3 in the main paper when we set the initial time $\tau = 0$.

$\blacksquare$

## B  ASODE algorithm

The architecture of our ASODE is presented in Figure 4 in our main paper and the process of ASODE is illustrated in Section 5.3. We transform them into ASODE algorithm 1.

**Algorithm 1** ASODE algorithm

---

**Input:** Training data $S := \{(\mathbf{x}_1, \mathbf{y}_1), \ldots, (\mathbf{x}_N, \mathbf{y}_N)\}$; parameters: $\alpha_1$, $\alpha_2$; evolution time: $T$; the number of samples drawn from the neighbor of $\mathbf{x}_n$: $K$; the radius of neighbourhood of $\mathbf{x}_n$: $\delta$; batch size $m$; number of batches $M$; number of epochs $T_1$, $T_2$; the loss $L_{ODE}$ and $L_{model}$; stepsize: $\eta_1$, $\eta_2$; an algorithm for generating adversarial samples: $AS(L, \mathbf{x})$.

**Initialization:** $\theta, \widetilde{\theta}$.

**for** epoch = 1 **to** $T_1$ **do**
  **for** mini-batch =1 **to** $M$ **do**
    Sample a mini-batch $\{(\mathbf{x}_n, y_n)\}_{n=1}^m$ from $S$
    **for** $i = 1$ **to** $m$ **do**
      sample $\mathbf{x}_i^{(1)}, \ldots, \mathbf{x}_i^{(K)}$ from $B_\delta(\mathbf{x}_i)$;
    **end for**
    Update $\theta = \theta - \eta_1 \frac{\partial L_{ODE}}{\partial \theta}$;
  **end for**
**end for**
**for** epoch = 1 **to** $T_2$ **do**
  **for** mini-batch =1 **to** $M$ **do**
    Sample a mini-batch $\{(\mathbf{x}_n, y_n)\}_{n=1}^m$ from $S$
    Update $\widetilde{\theta} = \widetilde{\theta} - \eta_2 \frac{\partial L_{model}}{\partial \theta}$;
  **end for**
**end for**
**Output:** $\theta, \widetilde{\theta}$.

---