# OpenReview forum: "Defending Against Adversarial Attacks via Neural Dynamic System"
_NeurIPS.cc/2022/Conference — NeurIPS 2022 Accept_

### Official Review · Reviewer_Ljok · 2022-06-23

**Rating:** 6
**Confidence:** 4
**Soundness:** 3 good
**Presentation:** 2 fair
**Contribution:** 2 fair

**Summary:**

This paper studies the neural network robustness issue from the perspective of dynamical systems.
The paper illustrates the drawbacks of previous works in this direction and proposes an asymptotically stable ordinary differential equation to overcome those drawbacks.

There are two major advantages of this proposed method:

1: ASODE considers asymptotically stable equilibrium points that can eliminate applied perturbations at initial conditions.

2: ASODE considers nonautonomous dynamical systems that are more general than previous works that focused on autonomous systems.

Numerical results have demonstrated promising improvements.

**Questions:**

I think the proposed ASODE has many overlaps with the SODEF method.
Could the authors itemize the major difference between ASODE and SODEF?

**Limitations:**

I did not find the discussion on the limitations of this proposed method.
Could the author provide a detailed discussion on the limitations here?

**Strengths And Weaknesses:**

Strengths: This paper considers robustness from the perspective of a well-developed control field, which helps to effectively connect those two fields.
The idea of using asymptotically stability rather than just stability can guarantee to eliminate of applied perturbations.
The numerical experiments provided in the paper are very promising, against all perturbations, the proposed ASODE has shown improvement over the previous state-of-the-art methods.

Weakness: I enjoyed reading this paper.
However, I found some grammar issues.
I suggest that authors carefully check the paper writing and improve the readability.

---

> ### Author Response · Authors · 2022-08-02
> **2-1 Answer to Reviewer Ljok**
>
> Thank you for your job in reviewing our paper. We are very sorry for the inconvenience caused by our presentations. To this end, following your comments, we correct our work in the revision and mark the corrections in red. All the references appear in the main paper of revision.
>
> *Question 1.* I think the proposed ASODE has many overlaps with the SODEF method. Could the authors itemize the major difference between ASODE and SODEF?
>
> *Answer:* We have discussed the difference between ASODE and SODEF in Section 1 of our main paper of revision (see lines 31-41).
>
> 1) In the papers [1,8,9], the general dynamical system explicitly depends on the argument $t$ and is referred to as a nonautonomous dynamical system. However, SODEF only considers the autonomous case, which is simply a special class of ODE. Our method uses the general dynamical system which explicitly depends on the argument $t$ and is referred to as a nonautonomous dynamical system.
>
> 2) As shown in Figure 1 (a), Lyapunov stability only controls the perturbations of the input rather than eliminating the effects of perturbations; simply maintaining the perturbations may still lead to misclassification of SODEF. Our method ASODE uses the asymptotic stability of nonautonomous dynamical system which is more strong than Lyapunov stability. If a clean instance is an asymptotically stable equilibrium point and the adversarial instance is in the neighborhood of this point, the asymptotic stability will reduce the adversarial noise to bring the adversarial instance close to the clean instance.
>
> 3) SODEF ensures that the output of the extractor is located in the neighborhood of the Lyapunov-stable equilibrium points; thus, the performance of SODEF depends on the extractor. Specifically, if the extracted features deviate from real features, SODEF would regard the false features as the Lyapunov-stable equilibrium points; accordingly, the perturbed input of neural ODE will converge to the incorrect features, which would lead to the misclassification of SODEF. ASODE can be simplified as follows: ODE + model. The ODE part can be regarded as a denoiser. If the input features deviate from real features, the ODE part of ASODE makes the perturbed instance close to the corresponding clean instance to avoid misclassification.

---

> ### Author Response · Authors · 2022-08-08
> **2-2 Answer to Reviewer Ljok**
>
>
>
> *Question 2.* I enjoyed reading this paper. However, I found some grammar issues. I suggest that authors carefully check the paper writing and improve the readability.
>
> *Answer:*  We have provided pseudo code and annotated equation (19) to improve the readability. These could be seen in the Answer to Questions 2 and 4 of ''Answer to Reviewer i5LX ''. We also emphasize our theoretical journey to fully satiate and improve readability, which can be seen in the Answer to ''Further improvement of the readability''  of Reviewer i5LX.

---

> ### Author Response · Authors · 2022-08-09
> **Discussion**
>
> Dear Reviewer Ljok, we will really appreciate it if the reviewer can go over our detailed response and revisions. Please feel free to ask us any questions you may still have and we will be more than happy to answer them. Thank you again for reviewing our paper and we look forward to discussing with you.

---

### Official Review · Reviewer_i5LX · 2022-06-29

**Rating:** 7
**Confidence:** 5
**Soundness:** 4 excellent
**Presentation:** 3 good
**Contribution:** 3 good

**Summary:**

The paper proposes a control theory approach to robustness by trying to turn "clean" data points into a kind of attractor basin, such that perturbations are "denoised" by being attracted back to better-behaved spaces of the neural network. This results in improved classification accuracy under strong attacks like PGD and AutoAttack

**Questions:**

The following reproducibility checklist questions have been incorrectly marked as "Not Applicable", please answer them:

* Did you describe the limitations of your work? [N/A] (You didn't, please address)
* Did you discuss any potential negative societal impacts of your work? [N/A]
* Did you include the code, data, and instructions needed to reproduce the main experimental results (either in the supplemental material or as a URL)? (You didn't, please address)
* Did you specify all the training details (e.g., data splits, hyperparameters, how they were chosen)? [Yes] (You did not, this is incorrect - e.g., optimizer details are missing)
* Did you report error bars (e.g., with respect to the random seed after running experiments multiple times)? [N/A]
* Did you include the total amount of compute and the type of resources used (e.g., type of GPUs, internal cluster, or cloud provider)? [N/A]

**Limitations:**

The approach is not applicable to any non-differentiable method like decision trees, and would be especially challenging to combine with current adversarial training techniques. These and other limitations should be addressed in the work.

**Strengths And Weaknesses:**

The idea of the paper is very interesting, and I like the approach. The improved robustness under attack is validating, my problems with the paper stem purely from its presentation. The paper does not state limitations, provides code, and is poorly organized for accessibility.

The paper can be hard to follow, e.g., Figure 2 is referenced before Figure 1 is even seen/referenced! In particular, there is a long theoretical journey to developing the proposed method that is very math-heavy. The work would benefit enormously from some pseudo code / a final "bring it all together" section. This is exacerbated by a lack of code and the authors ignoring/lying the reproducibility checklist (see questions), and I am strongly against an essentially unanotated equation (19) being the final distillation - its needlessly unreadable.


I overall like the idea, but believe two key related works are missing from the discussion. Most importantly [A], which is the first work to use control theory and dynamic systems to improve robustness, showing global perturbation bounds for a neural network by treating the parameters as an evolving system that needs to be kept stable. Second [B] showed that RBF style functions are naturally more robust to attack, of which I think this work is related by using the ODE to create attractor basins near "clean" samples is reminiscent at a high level.

[A] Rahnama, A., Nguyen, A. T., & Raff, E. (2020). Robust Design of Deep Neural Networks against Adversarial Attacks based on Lyapunov Theory. The IEEE/CVF Conference on Computer Vision and Pattern Recognition (CVPR), 8178–8187. http://arxiv.org/abs/1911.04636
[B] Biggio, B., Fumera, G., & Roli, F. (2014). Security evaluation of pattern classifiers under attack. IEEE Transactions on Knowledge and Data Engineering, 26(4), 984–996. https://doi.org/10.1109/TKDE.2013.57

-------------

I was in a bad mood when I wrote the OG review, I again apologize. I think the revision has satisficed - though not fully satiated, improved readability. But I need to draw the line between what I would prefer vs what is indeed an improvement and valuable contribution. I've raised my score accordingly. If the authors feel they can further improve the presentation I'm willing to further raise if successful.

----

Re-reading I've updated my score. I'm having trouble separating my personal style preferences that are quite different from the reviews - and style differences are no reason to reject a work. So I'm hedging that others will find the paper more inviting given the improvements than I do. I think all objectively 'bad' writing style has been fixed.

---

> ### Author Response · Authors · 2022-08-02
> **4-1 Answer to Reviewer i5LX**
>
> Thank you for your job in reviewing our paper. We are very sorry for the inconvenience caused by our presentations. To this end, following your comments, we correct our work in the revision and mark the corrections in red. All the references appear in the main paper of revision.
>
> *Question 1.* My problems with the paper stem purely from its presentation. The paper does not state limitations, provides code, and is poorly organized for accessibility.
>
> *Answer:* For limitation: The limitation of our work is that ASODE couldn't be applied to safety-critical fields directly. It is imperative to study how to apply ASODE in real-world applications.
>
> For code: We have strict protocols for code release as this research is partially funded by corporate funding. We will upload the training code as soon as permission is granted.
>
> For organization: In Section 1, we introduce that DNN can be significantly affected by human-imperceptible perturbations and dynamical systems can be used to defend against adversarial attacks. Therefore, we use asymptotic stability of the general nonautonomous dynamical system to reduce the perturbation on the input. In Section 2 and 3, we list relevant works and basic conceptions to make preparation. From Section 4 to 5, we linearize the nonautonomous ODE and place constraints on its corresponding linearization to make all clean instances act as its asymptotically stable equilibrium points. In Section 6, we do experiments to verify our method.
>
> The answers to limitations and codes are added in the checklist of the main paper of revision (see lines 411-414 and 423-426).
>
> *Question 2.*
> The work would benefit enormously from some pseudo code / a final "bring it all together" section. This is exacerbated by a lack of code and the authors ignoring/lying the reproducibility checklist (see questions), and I am strongly against an essentially unanotated equation (19) being the final distillation - its needlessly unreadable.
>
> *Answer:* The architecture of our ASODE is presented in Figure 4 of our main paper and the process of ASODE algorithm is illustrated in Section 5.3 of our main paper. We transform them into the pseudo code in below ASODE algorithm: For simplicity, we denote the loss (19) in our main paper as $L_{ODE}$ and denote the loss (20) in our main paper as $L_{model}$. We add the pseudo code in the Appendix C of the revision (see lines 104-107) and emphasize it in the main paper of the revision (see lines 256-257).
>
> | ASODE algorithm ||        |       |       |       |
> |:--------|:-----:|:------:|:-----:|:-----:|:-----:|
> |$Input:$   Training data $S:= { (\mathbf{x} _1, \mathbf{y} _1), \ldots, (\mathbf{x} _N, \mathbf{y} _N)}$; parameters: $\alpha _1$, $\alpha _2$; evolution time: $T$; the number of samples drawn from the neighbor of $\mathbf{x} _n$:  $K$; the radius of neighbourhood of $\mathbf{x} _n$: $\delta$; batch size $m$; number of batches $M$; number of epochs $T _1$, $T _2$;  the loss $L _{ODE}$ and $L _{model}$; stepsize: $\eta _1$, $\eta _2$; an algorithm for generating adversarial samples: $AS(L,\mathbf{x})$.
>  |  |   |  |  |  |
> |$Initialization:$ $\theta$, $\widetilde{\theta}$.
>  |  |   |  |  |  |
> $\mathbf{for}$ epoch=1 to $T _1$ $\mathbf{do}$
> &emsp;&emsp;$\mathbf{for}$ mini-batch=1 to $M$ $\mathbf{do}$
> &emsp;&emsp;&emsp;&emsp;Sample a mini-batch $(\mathbf{x} _n, y _n) _{n=1}^{m}$ from $S$;
> &emsp;&emsp;&emsp;&emsp;$\mathbf{for}$ $i=1$ to $m$ $\mathbf{do}$
> &emsp;&emsp;&emsp;&emsp;&emsp;&emsp;&emsp;&emsp;sample $\mathbf{x} _i^{(1)}, \ldots, \mathbf{x} _i^{(K)}$ from $B _\delta(\mathbf{x} _i)$;
> &emsp;&emsp;&emsp;&emsp;&emsp;&emsp;&emsp;&emsp;Obtain adversarial data $\mathbf{x} _i^*=AS(L _{ODE}, \mathbf{x} _i)$;
> &emsp;&emsp;&emsp;&emsp;$\mathbf{end}$ $\mathbf{for}$
> &emsp;&emsp;&emsp;&emsp;Update $\theta=\theta-\eta _1\frac{\partial L _{ODE}}{\partial \theta}$;
> &emsp;&emsp;$\mathbf{end}$ $\mathbf{for}$
> $\mathbf{end}$ $\mathbf{for}$
> | $\mathbf{for}$ epoch=1 to $T _2$ $\mathbf{do}$
> &emsp;&emsp;$\mathbf{for}$ mini-batch=1 to $M$ $\mathbf{do}$
> &emsp;&emsp;&emsp;&emsp;Sample a mini-batch $\{(\mathbf{x} _n, y _n)\} _{n=1}^{m}$ from $S$;
> &emsp;&emsp;&emsp;&emsp;$\mathbf{for}$ $i=1$ to $m$ $\mathbf{do}$
> &emsp;&emsp;&emsp;&emsp;&emsp;&emsp;&emsp;&emsp;Obtain adversarial data $\mathbf{x} _i^*=AS(L _{model}, \mathbf{x} _i);$
> &emsp;&emsp;&emsp;&emsp;$\mathbf{end}$ $\mathbf{for}$
> &emsp;&emsp;&emsp;&emsp;Update $\widetilde{\theta}=\widetilde{\theta}-\eta _2\frac{\partial L _{model}}{\partial \widetilde{\theta}}$;
> &emsp;&emsp;$\mathbf{end}$ $\mathbf{for}$
> $\mathbf{end}$ $\mathbf{for}$
> |  |  |  |  |
> |$Output:$ $\theta$, $\widetilde{\theta}$.

---

> > ### Comment · Reviewer_i5LX · 2022-08-04
> > **Pseudo code in Markdown!?**
> >
> > Color me impressed at the markdown in open review! Thats above and beyond.
> >
> > For your pseudo-code, equation 20 you have defined as:
> >
> > $V(\mathbf{z}, \tau)=\int_{0}^{+\infty}\left\|\mathbf{s}_{\tau}\left(\mathbf{z}-\mathbf{x}^{*}, 0, t\right)\right\|^{p} d t$
> >
> > I think that is highly non-standard, and needs some further explanation on how its evaluation is implemented. Do you do numerical integration of this? Do you hard-code the derivative of the integration of let auto-diff of numerical integration handle it? There are still a number of questions.
> >
> > Equation (19) I understand as being the left-hand-side as what is evaluated, but could be made explicit given the equation has an $=$ in it.
> >
> > I apologize my initial review was a bit hard necessarily, and that I'm being a bit brief. I have some external factors sapping my brain of sleep at the moment. Your rebuttal has resolved the majority of my other issues to the point many I would consider stylistic differences.

---

> > > ### Author Response · Authors · 2022-08-04
> > > **Answer to Reviewer i5LX for "Pseudo code in Markdown!?"**
> > >
> > > Thank you for your job in reviewing our answers. We are really sorry for the inconvenience caused by our presentations. To this end, following your comments, we correct our work and update the revision. The corrections in the version is marked in red. All the references and formula numbers appear in the main paper and appendix of revision.
> > >
> > > *Question 1.* Pseudo code in Markdown!?
> > >
> > > *Answer:* Limited by the compiler of the OpenReview, we make the pseudo code by tables and texts in Markdown.
> > >
> > > *Question 2.* For your pseudo-code, equation 20 you have defined as:
> > >
> > > $$
> > > V(\mathbf{z}, t)=\int _0^{+\infty}||\mathbf{s} _t(\mathbf{z}-\mathbf{x}^*,0,\tau)||^pd\tau, (20)
> > > $$
> > >
> > > I think that is highly non-standard, and needs some further explanation on how its evaluation is implemented. Do you do numerical integration of this? Do you hard-code the derivative of the integration of let auto-diff of numerical integration handle it? There are still a number of questions.
> > >
> > > *Answer:*
> > > The equation (20) (see lines 86-87 in Appendix A.3) you mentioned is in the proof of Theorem 3, not in our pseudo-code. The equation (20) is used to prove Theorem 3 in our main paper.
> > >
> > > $$
> > > V(\mathbf{z}, t)=\int _0^{+\infty}||\mathbf{s} _t(\mathbf{z}-\mathbf{x}^*,0,\tau)||^pd\tau, (20)
> > > $$
> > >
> > > In the proof of Theorem 3, we get (22) (see lines 89 in Appendix A.3):
> > >
> > > $$
> > > \dot{V}(\mathbf{z}-\mathbf{x}^*, \tau)=\frac{\partial V(\mathbf{z}-\mathbf{x}^*, \tau)}{\partial \mathbf{z}}\mathbf{h}(\mathbf{z}-\mathbf{x}^*, \tau)+\frac{\partial V(\mathbf{z}-\mathbf{x}^*, \tau)}{\partial \tau}
> > > =\frac{\partial V(\mathbf{z}-\mathbf{x}^*, \tau)}{\partial \tau}-||\mathbf{z}-\mathbf{x}^*||^p. (22)
> > > $$
> > >
> > > It only remains to estimate $\frac{\partial V(\mathbf{z}, \tau)}{\partial \tau}$, let $\gamma:= \frac{p}{2}$, then, from (20),
> > >
> > > $$
> > > \frac{\partial V(\mathbf{z}-\mathbf{x}^*, \tau)}{\partial \tau}
> > > =\int_0^{+\infty} \frac{\partial[\mathbf{s}^{\top}_\tau(\mathbf{z}-\mathbf{x}^*,0,t)\mathbf{s}_\tau(\mathbf{z}-\mathbf{x}^*,0,t)]^\gamma}{\partial \tau}dt
> > > =\int_0^{+\infty} 2\gamma[\mathbf{s}^{\top}_\tau(\mathbf{z}-\mathbf{x}^*,0,t)\mathbf{s}_\tau(\mathbf{z}-\mathbf{x}^*,0,t)]^{\gamma-1}\mathbf{s}^{\top}_\tau(\mathbf{z}-\mathbf{x}^*,0,t)\frac{\partial\mathbf{s}_\tau(\mathbf{z},0,t)}{\partial \tau}dt|\frac{\partial V(\mathbf{z}-\mathbf{x}^*, \tau)}{\partial \tau}|\leq \int_0^{+\infty}2\gamma||\mathbf{s}_\tau(\mathbf{z}-\mathbf{x}^*,0,t)||^{\gamma-1}||\frac{\partial\mathbf{s}_\tau(\mathbf{z}-\mathbf{x}^*,0,t)}{\partial \tau}||dt.
> > > $$
> > > We then bound $\frac{\partial V(\mathbf{z}-\mathbf{x}^*, \tau)}{\partial \tau}$ and get (23) from (22)
> > >
> > > $$
> > > \dot{V}(\mathbf{z}-\mathbf{x}^*, t)\leq -(1-m)||\mathbf{z}-\mathbf{x}^*||^p.(23)
> > > $$
> > >
> > > Then, (21) （see lines 87-89 in Appendix A.3） and (23) show that $V$ is a suitable Lyapunov function for applying the Lemma 5 (see lines 61-66 in Appendix A.3) to conclude the exponential stability.
> > >
> > > In summary, formulation (20) (see lines 86-87 in Appendix A.3) is only used to prove Theorem 3 in our main paper and we don't use its numerical integration. Therefore,
> > > we do not need to evaluate (20).
> > >
> > > *Question 3.* Equation (19) I understand as being the left-hand-side as what is evaluated, but could be made explicit given the equation has an ``$=$'' in it.
> > >
> > > *Answer:*
> > > Thank you for your suggestions. Again, we are very sorry for the inconvenience caused by our presentations.
> > > Following your suggestions, we denote the empirical loss of ODE part of our ASODE as $L_{ODE}$ and make $L_{ODE}$ be the left-hand-side of (19) in the main paper as what is evaluated. Then, we add ``$=$'' in (19) of our main paper of revision (see lines 248-249). We present it here:
> > >
> > >  $$
> > > L_{ODE}=\mathop{min}\limits _{\theta}\frac{1}{N}\sum\limits _{n=1}^{N}\lbrace\Phi(\mathbf{s}(\mathbf{x}
> > >  _n,T),\mathbf{x} _n)+\frac{1}{K}\sum\limits _{k=1}^{K}\Phi(\mathbf{s}(\mathbf{x} _n^{(k)},T),\mathbf{x} _n)+
> > >     \frac{1}{T}\sum\limits _{\tau=0}^{T-1}[
> > >     \alpha _1(||\mathbf{h} _\theta(\mathbf{x} _n, \tau)||
> > >     +||(\frac{\partial\mathbf{h}(\mathbf{s}(\mathbf{x} _n,t), t)}{\partial t}) _{t=\tau}||/||\mathbf{s}(\mathbf{x} _n,\tau)||)+\alpha _2(\exp{(\sum\limits _{i=1}^{d}-|\nabla\mathbf{h} _\theta(\mathbf{x} _n,\tau)] _{ii}|)}
> > >     + \exp{(\sum\limits _{i=1}^{d}(-|\nabla\mathbf{h} _\theta(\mathbf{x} _n,\tau)] _{ii}|
> > >     +\sum\limits _{j=1,j\neq i}^{d}|\nabla\mathbf{h} _\theta(\mathbf{x} _n,\tau)] _{ij}|)))}]\rbrace.(19)
> > > $$

---

> ### Author Response · Authors · 2022-08-02
> **4-2 Answer to Reviewer i5LX**
>
>
> *Continue  Answer to Question 2:*
>
> Equation (19) in the main paper is an empirical Lagrangian of optimization problem (14)-(18) of our main paper. Equation (19) is annotated in Section 5.3 and is elaborated in Appendix B of revision (see lines 108-112).  We emphasize them below.
> $$
> \min _{\theta}  \underset{{\mathbf{x} \sim \mathcal{D} _\mathbf{x}}}{\mathbb{E}} \underset{ \widetilde{\mathbf{x}} \sim \mathcal{D} _\widetilde{\mathbf{x}}}{\mathbb{E}}[I _{B _\delta(\mathbf{x})}(\widetilde {\mathbf{x}})\Phi(\mathbf{z}(T),\mathbf{x})],(4)
> $$
> s.t.
>
> $$
> \mathbb{E}[\mathbf{h} _\theta(\mathbf{x},t)]\leq\epsilon _1, (5)
> $$
> $$
> \mathbb{E}\left[||\frac{\partial\mathbf{h} _\theta(\mathbf{z}, t)}{\partial t}||\big/||\mathbf{z}||\right]\leq\epsilon _2, (6)
> $$
>
> $$
> \mathbb{E}[[\nabla\mathbf{h} _\theta (\mathbf{x},t)] _{ii}]< 0, (7)
> $$
>
> $$
> \mathbb{E}\left[|[\mathbf{h} _\theta(\mathbf{x},t)] _{ii}|-\sum\limits _{j\neq i}|[\nabla\mathbf{h} _\theta(\mathbf{x},t)] _{ij}|\right] > 0,(8)
> $$
> $$
> \forall t\in [0,T]\quad and\quad i,j=1,2,\cdots,d,
> $$
>
>  According to the optimization problem (4), the constraints (5), (6), (7) and (8) correspond to the following discretization forms (11), (12}), (13) and (14) respectively.  Furthermore, the objective (4) corresponds to the following (9) and (10). Based on the analysis of the optimization problem (4), we get that the smaller the following values (9)- (14) are, the better the neural ODE is. Therefore, we construct the discrete objective in the main paper.
>
> $$
> \frac{1}{N}\sum\limits _{n=1}^{N}\Phi(\mathbf{s}(\mathbf{x} _n,T),\mathbf{x} _n),(9)
> $$
>
> $$
> \frac{1}{N K}\sum\limits _{n=1}^{N}\sum\limits _{k=1}^{K}\Phi(\mathbf{s}(\mathbf{x} _n^{(k)},T),\mathbf{x} _n),(10)
> $$
>
> $$
> \frac{1}{N T}\sum\limits _{n=1}^{N}\sum\limits _{\tau=0}^{T-1}||\mathbf{h} _\theta(\mathbf{x} _n, \tau)||,(11)
> $$
>
> $$
> \frac{1}{N T}\sum\limits_{n=1}^{N}\sum\limits _{\tau=0}^{T-1}||\big(\frac{\partial\mathbf{h}(\mathbf{s}(\mathbf{x} _n,t), t)}{\partial t}\big) _{t=\tau}||\bigg/||\mathbf{s}(\mathbf{x} _n,\tau)||,(12)
> $$
>
> $$
> \frac{1}{N T}\sum\limits _{n=1}^{N}\sum\limits _{\tau=0}^{T-1}\sum\limits _{i=1}^{d}-|\nabla\mathbf{h} _\theta(\mathbf{x}_n,\tau)] _{ii}|,(13)
> $$
>
> $$
> \frac{1}{N T}\sum\limits _{n=1}^{N}\sum\limits _{\tau=0}^{T-1}\sum\limits _{i=1}^{d}\bigg(-|\nabla\mathbf{h} _\theta(\mathbf{x} _n,\tau)] _{ii}|
>     +\sum\limits _{j=1,j\neq i}^{d}|\nabla\mathbf{h} _\theta(\mathbf{x} _n,\tau)] _{ij}|\bigg),(14)
> $$
> Based on the above analysis, we make the following equation (19) in the main paper
>
>  $$
> \mathop{min}\limits _{\theta}\frac{1}{N}\sum\limits _{n=1}^{N}\lbrace\Phi(\mathbf{s}(\mathbf{x}
>  _n,T),\mathbf{x} _n)+\frac{1}{K}\sum\limits _{k=1}^{K}\Phi(\mathbf{s}(\mathbf{x} _n^{(k)},T),\mathbf{x} _n)+
>     \frac{1}{T}\sum\limits _{\tau=0}^{T-1}[
>     \alpha _1(||\mathbf{h} _\theta(\mathbf{x} _n, \tau)||
>     +||(\frac{\partial\mathbf{h}(\mathbf{s}(\mathbf{x} _n,t), t)}{\partial t}) _{t=\tau}||/||\mathbf{s}(\mathbf{x} _n,\tau)||)+\alpha _2(\exp{(\sum\limits _{i=1}^{d}-|\nabla\mathbf{h} _\theta(\mathbf{x} _n,\tau)] _{ii}|)}
>     + \exp{(\sum\limits _{i=1}^{d}(-|\nabla\mathbf{h} _\theta(\mathbf{x} _n,\tau)] _{ii}|
>     +\sum\limits _{j=1,j\neq i}^{d}|\nabla\mathbf{h} _\theta(\mathbf{x} _n,\tau)] _{ij}|)))}
>     ]\rbrace
> $$

---

> ### Author Response · Authors · 2022-08-02
> **4-3 Answer to Reviewer i5LX**
>
> *Question 3.* I overall like the idea, but believe two key related works are missing from the discussion. Most importantly [A], which is the first work to use control theory and dynamic systems to improve robustness, showing global perturbation bounds for a neural network by treating the parameters as an evolving system that needs to be kept stable. Second [B] showed that RBF style functions are naturally more robust to attack, of which I think this work is related by using the ODE to create attractor basins near "clean" samples is reminiscent at a high level.
>
> [A] Rahnama, A., Nguyen, A. T., \& Raff, E. (2020). Robust Design of Deep Neural Networks against Adversarial Attacks based on Lyapunov Theory. The IEEE/CVF Conference on Computer Vision and Pattern Recognition (CVPR), 8178–8187.
>
> [B] Biggio, B., Fumera, G., \& Roli, F. (2014). Security evaluation of pattern classifiers under attack. IEEE Transactions on Knowledge and Data Engineering, 26(4), 984–996.
>
> *Answer:*
> Thank you for you suggestion and we add the discussion of [A] and [B] in Section 2 of the main paper of revision (see lines 90-92).
>
> [A] is the first work to use control theory and dynamic systems to improve robustness, showing global perturbation bounds for a neural network by treating the parameters as an evolving system that needs to be kept stable. The difference between [A] and our works are follows:
>
> 1). [A] treats each individual layer of the DNN as a nonlinear dynamical system, while our dynamic system is not. Our works ASODE can be simplified as follows: ODE + model. The ODE part can be regarded as a denoiser, which helps to reduce the noise, including the adversarial noise, to rectify the perturbed instance.
>
> 2). [A] follows the general definition for the nonlinear system $H$,
>
> $$H : \dot{x}=f(x,u),y=h(x,u),(15)$$
>
> where $u$ is the input, $x$ is the state which corresponds to $\theta$ and $\widetilde{\theta}$ in our work. The nonlinear mappings $f$ and $h$ model the relationship among the input signal $u$, the internal states of the system $x$ and the output signal $y$.
>
> However, we follows the nonautonomous system (16) and (17),
> $$\frac{d\mathbf{z}(t)}{dt} = \mathbf{h}_\theta(\mathbf{z}(t),t), \forall t\in[0,T],(16)$$
>
> $$\mathbf{y}=\mathbf{f}_{\widetilde{\theta}}(\mathbf{z}(T)),\mathbf{z}(0)=\mathbf{x}, \mathbf{x}\in\mathcal{X},(17)$$
>
> where $\mathbf{z}(t)$ is the input corresponding to $u$ in (15), but $t$ is time which doesn't correspond to the state. Meanwhile, Our nonlinear mappings $f$ and $h$ are dependent on the state, which is largely different from the definition in (15).
>
> 3). The motivation of [A] is that if a nonlinear system is stable and robust, its output signals are close, when their respective input signals are also close enough in the Euclidean space. Absolutely, the approach in [A] just controls the error caused by the perturbation on the input and it couldn't eliminate the effects of perturbations. However, our motivation is to use the asymptotic stability to reduce perturbations on the initial point. If a clean instance is an asymptotically stable equilibrium point and the adversarial instance is in the neighborhood of this point, the asymptotic stability will reduce the adversarial noise to bring the adversarial instance close to the clean instance.
>
> For [B], the reviewer think that RBF style functions are naturally more robust to attack and its work related by using the ODE to create attractor basins near "clean" samples is reminiscent at a high level. The authors of [B] give the following description in Section 4.3:
>
> ``The attack samples can be considered outliers with respect to the legitimate training samples,and, for large values of the RBF kernel, the SVM discriminant function tends to overfit, forming a “peak” around each individual training sample. Thus, it exhibits relatively high values also in the region of the feature space where the attack samples lie, and this allows many of the corresponding testing intrusions. Conversely, this is not true for lower values of the RBF kernel, where the higher spread of the RBF kernel leads to a smoother discriminant function, which exhibits much lower values for the attack samples.''
>
> Therefore, we can't get that RBF style functions are naturally more robust to attack, but RBF style functions with lower values are more robust to attack. RBF kernel with large values forms a “peak” around each individual training sample and allows many of the corresponding testing intrusions. Besides, there is no explaination about using ODE to create attractor basins near "clean" samples in [B]. The authors of [B] uses the property of radial basis function (RBF) to improve the robust, not ODE.

---

> ### Author Response · Authors · 2022-08-03
> **4-4 Answer to Reviewer i5LX**
>
> *Question 4.* The following reproducibility checklist questions have been incorrectly marked as "Not Applicable", please answer them:
>
> *Answer:* Thank you for your comments. We answer them in the checklist of the revision (see lines 411-439).
>
> 1). Did you describe the limitations of your work? [N/A] (You didn't, please address)
>
> *Answer:*
> The limitation of our work is that ASODE couldn't be applied to safety-critical fields directly.
>
> 2). Did you discuss any potential negative societal impacts of your work? [N/A]
>
> *Answer:*
> Our works has no potential negative societal impacts.
>
>
> 3). Did you include the code, data, and instructions needed to reproduce the main experimental results (either in the supplemental material or as a URL)? (You didn't, please address)
>
> *Answer:*
> We have strict protocols for code release as this research is partially funded by corporate funding. We will upload the training code as soon as permission is granted.
>
>
> 4). Did you specify all the training details (e.g., data splits, hyperparameters, how they were chosen)? [Yes] (You did not, this is incorrect - e.g., optimizer details are missing)
>
> *Answer:*
> Thanks for your suggestion. According to the official pytorch encapsulated dataset,for MINIST,  CIFAR-10 and CIFAR-100, we use 50000 images for training dataset and 10000  images for test dataset. We use a batch size of 128 for training. For optimizer, we use SGD with momentum=$0.9$, learning rate=$0.1$ and weight decay $5e-4$.
>
> 5). Did you report error bars (e.g., with respect to the random seed after running experiments multiple times)? [N/A]
>
> *Answer:*
> We didn't report error bars. As models take a significant amount of time to train, averaging over many seeds would be computationally impractical.
>
> 6). Did you include the total amount of compute and the type of resources used (e.g., type of GPUs, internal cluster, or cloud provider)? [N/A]
>
> *Answer:*
> We use a internal cluster with 8 NVIDIA GeForce RTX 2080Ti.

---

> ### Author Response · Authors · 2022-08-07
> **Further improvement of the readability**
>
> *Question:* I think the revision has satisficed - though not fully satiated, improved readability.
>
> *Answer:* Dear Reviewer i5LX, thank you for your comments. Following your comments, we correct our work and update the revision. The corrections in the revision are marked in red. All formula numbers appear in the main paper of revision. To fully satiate and improve readability, we make the following revision.
>
> *Revision 1:* We change the positions of Figure 1 and Figure 2 to make Figure 1 referenced before Figure 2 in the main paper of our revision (see lines 35-36 and 55-56). Meanwhile, we rewrite the name of the corresponding figures in the main paper of our revision (see lines 34, 43, 62, 114, 117, 138, 147, and 148).
>
> *Revision 2:* The theoretical journey to developing the proposed method has been illustrated in Sections 4 and 5 (see lines 164-206). We emphasize and summarize the theoretical journey: To sum up, we linearize the slowly varying nonautonomous ODE (1) and impose constraints on its linearization (9) to make all clean instances be asymptotically stable equilibrium points of (1). We add this summary at the end of Section 4 in the main paper of our revision (see lines 207-208 ).
>
> We have tried our best to improve the readability. Would you please provide us suggestions to further improve the readability, if possible? Thanks for your time.

---

> ### Author Response · Authors · 2022-08-09
> **Thanks**
>
> Thank you for your comments and raising the rating.

---

### Official Review · Reviewer_v7EB · 2022-07-06

**Rating:** 4
**Confidence:** 2
**Soundness:** 3 good
**Presentation:** 3 good
**Contribution:** 3 good

**Summary:**

The paper designs an ordinary differential equation (ODE) that can regularize adversarial training samples to be close to clean samples. The ODE is developed based on the asymptotic stability achieved by a slowly time-varying dynamic system. The evaluation involves different existing attacking methods and datasets with ResNet18. The experimental results show that the proposed ASODE can defend against existing attacks and outperforms SOTA defense methods.


**Questions:**

Line 269, why the model $\textbf{f}_{\widetilde{\theta}}$ is only trained for 100 epochs?


**Limitations:**

**Writing issues**:

It would be better to introduce the meaning of a symbol at its first appearance.

**Strengths And Weaknesses:**


**Strength**

++ It is an interesting idea to mitigate noise and perturbation of adversarial attacks from the perspective of dynamic systems.

++ The paper presents a detailed theoretical analysis of how a perturbed instance is optimized to converge to a clean instance.



**Weakness**

-- Lack of details of attack methods and their settings

-- Missing ablation study of ASODE

-- Missing sensitivity analysis of critical parameters of ASODE, such as the choice of $\delta$

---

> ### Author Response · Authors · 2022-08-02
> **Answer to Reviewer v7EB**
>
> Thank you for your job in reviewing our paper. We are very sorry for the inconvenience caused by our presentations. To this end, following your comments, we correct our work in the revision and mark the corrections in red. All the references appear in the main paper of revision.
>
> *Question 1.* Lack of details of attack methods and their settings.
>
> *Answer:*  Some settings of attack methods have been showed in Section 6.1 of the main paper of revision (see lines 266-269). Following your comments, we have supplemented our details of attack methods in appendix D of the revision (see lines 108-112). Besides, we emphasize the supplementary details in the main paper of revision (see line 269). According to the same experimental settings as for SODEF [7], for two vanilla white-box FGSM and PGD attacks, we use $L_\infty$ norm with maximum perturbation $\epsilon = 0.3$ for MNIST and $\epsilon= 0.031$ for CIFAR-10 and CIFAR-100. We iterate PGD 20 times with step size $0.007$. For the ensemble AutoAttack, we use the standard $L_2$ norm with maximum perturbation $\epsilon = 0.5$ as in the benchmark.
>
> *Question 2.* Missing Ablation Study of ASODE.
>
> *Answer:* We have done experiments about the ablation study of ASODE and the results can be seen in Tables 2-4 in the main paper. The ``NO ODE'' in the tables corresponds to ASODE without the ODE part. From Tables 2-4 in the main paper, We find that ASODE is more robust than NO ODE, which indicates that ODE helps improve the robustness. We emphasize the experimental results of ablation in the main paper of the revision (see lines 307-309).
>
> *Question 3.*  Missing sensitivity analysis of critical parameters of ASODE, such as the choice of $\delta$.
>
> *Answer:* Following your comments, we have supplemented the sensitivity analysis of $\delta$ and the results are presented in Table 1.
>
> Table 1:
> | $\delta$ |  0.01 |  0.02  |  0.03 |  0.04 |  0.05 |
> |:--------:|:-----:|:------:|:-----:|:-----:|:-----:|
> |   None   | 95.08 |  95.12 | 95.06 | 95.02 | 95.00 |
> |   FGSM   | 41.86 | 52.18  | 69.56 | 70.89 | 72.24 |
> |    PGD   | 36.18 |  45.95 | 57.15 | 57.84 | 58.68 |
>
> Table 1 shows, the classification accuracy increases as $\delta$ increases. We add this sensitivity analysis of $\delta$ in Appendix F of the revision (see lines 116-118) and we emphasize it in the main paper of revision (see lines 278).
>
>
> *Question 4.*
> Line 269, why the model $f_{\widetilde{\theta}}$ is only trained for 100 epochs?
>
> *Answer:*
> As shown in Figure F1 in Appendix E of the revision (see lines 113-115), during the training of ASODE, the accuracy converges when $f_{\widetilde{\theta}}$ is trained for 100 epochs. We emphasize the conclusion in the main paper of the revision (see line 276).

---

> ### Author Response · Authors · 2022-08-06
> **Discussion**
>
> Dear Reviewer v7EB, we will really appreciate it if the reviewer can go over our detailed response and revisions. Please feel free to ask us any questions you may still have and we will be more than happy to answer them.  Thank you again for reviewing our paper and we look forward to discussing with you.

---

> ### Author Response · Authors · 2022-08-08
> **Discussion**
>
> Dear Reviewer v7EB, since your rating is low, we hope to have more discussion with you. If you have any other questions, please let us know. Many thanks for your time.

---

### Official Review · Reviewer_7GZb · 2022-07-09

**Rating:** 8
**Confidence:** 5
**Soundness:** 3 good
**Presentation:** 4 excellent
**Contribution:** 3 good

**Summary:**

Inspired by the asymptotic stability of the general nonautonomous dynamical system, the asymptotic stability will reduce the adversarial noise to bring the adversarial instance close to the clean instance. Motivated by the theoretical results, the authors propose a nonautonomous neural ordinary differential equation (ASODE) and place constraints on its corresponding linear time-variant system to make all clean instances act as the asymptotically stable equilibrium points of a slowly time-varying system. In the implementation, the authors convert the constraints to regularizers of the loss function. The experimental results show that ASODE improves robustness against adversarial attacks and outperforms the state-of-the-art methods.

**Questions:**

See the weaknesses above.

**Limitations:**

The paper has no potential negative societal impact.

**Strengths And Weaknesses:**

Strengths:
1. The paper bridges the gap between the asymptotic stability of ODE and robustness against adversarial attacks. The asymptotic stability of ODE helps reduce the adversarial noise.

2. The authors provide some theories to transform the general nonautonomous dynamical system into a linear time-variant system. Based on the transformation, the authors create a new loss function by the constraints on the linear time-variant system to make all clean instances act as the asymptotically stable equilibrium points.

3. The authors carefully proved the claims and further propose a nonautonomous neural ordinary differential equation (ASODE) that uses the asymptotic stability of ODE to bring the adversarial instance close to the clean instance. Experiments show that ASODE improves robustness against adversarial attacks and outperforms the state-of-the-art methods, which shows the significance of this work.

4. The analysis presented in the main paper and the appendix is thorough enough.

Weaknesses:

1. the reason why the authors use a dynamic system in this paper is not elaborated on.

2. The application of theorem 2 and theorem 3 is unclear. What is the relationship between theorem 2 and theorem 3?

---

> ### Author Response · Authors · 2022-08-02
> **Answer to Reviewer 7GZb**
>
> Thank you for your job in reviewing our paper. We are very sorry for the inconvenience caused by our presentations. To this end, following your comments, we clarify your queries below. All the references appear in the main paper of revision.
>
> *Question 1.* The reason why the authors use a dynamic system in this paper is not elaborated on.
>
> *Answer:* The reasons why we use a dynamic system in this paper are discussed in the Abstract and Introduction (see lines 3-11 and 20-29). The performance of DNN can be significantly affected by human-imperceptible perturbations that can drastically change the output of network [4,5]. Recently, some methods have been proposed to defend against adversarial attacks from the perspective of dynamic systems. The paper [6] proposes a time-invariant steady neural ODE (TisODE) to limit the evolution of the curves by forcing the integrand to be close to zero. However, this approach does not guarantee that small perturbations of the initial point will lead to small perturbations in the output at time $T$. Inspired by the asymptotic stability of the general nonautonomous dynamical system, we find that if a clean instance is an asymptotically stable equilibrium point and the adversarial instance is in the neighborhood of this point, the asymptotic stability will reduce the adversarial noise to bring the adversarial instance close to the clean instance. Therefore, we propose to make each clean instance be the asymptotically stable equilibrium points of a slowly time-varying dynamic system in order to defend against adversarial attacks.
>
> *Question 2.* The application of Theorem 2 and Theorem 3 is unclear. What is the relationship between Theorem 2 and Theorem 3?
>
> *Answer:* The application and relationship of Theorem 2 and Theorem 3 are discussed in Section 4 (see lines 164-194).
> $$
> \frac{d\mathbf{z}(t)}{dt} = \mathbf{h}(\mathbf{z}(t),t).   (1)
> $$
>
> $$
>  \frac{d\mathbf{z}(t)}{dt}=\mathbf{A}(t)(\mathbf{z}-\mathbf{x}^*). (2)
> $$
>
>
> Theorem 2 linearizes ODE (1) to (2) using the Lyapunov linearization method. To make all equilibrium points of (2) asymptotically stable, we deduce the stability of the nonautonomous system (1) by studying only the "frozen" systems; that is, the ODE (1) with time "frozen" at $r$. In other words, if $r\in\mathbb{R}^+$ is any fixed number, we can think of the autonomous system
>
> $$
>  \frac{d\mathbf{z}(t)}{dt} = \mathbf{h}(\mathbf{z}(t),r), \forall t\geq 0. (3)
> $$
>
> If $\mathbf{x}^*$ is the equilibrium point, then $\mathbf{A}=[\frac{\partial{\mathbf{h}(\mathbf{z},r)}}{\partial\mathbf{z}}]_{\mathbf{z}=\mathbf{x}^*}$ is a constant matrix and the linearization (2) becomes $\dot{\mathbf{z}}(t)=\mathbf{A}(\mathbf{z}-\mathbf{x}^*)$. However, even if each of the frozen systems (3) is exponentially stable, the overall system (1) can be unstable [10]. Theorem 3 proves that if each frozen system is exponentially stable and the system is slowly varying, the overall system is indeed exponentially stable. Based on Theorems 2 and 3, we impose constraints on ODE (1) to be a slowly varying systems and linearize it to be (2). We then impose constraints on the linearization (2) to make the ODE (1) asymptotically stable.

---

### Meta-Review · Area_Chair_bsP9 · 2022-08-24

**Recommendation:** Accept
**Confidence:** Certain

**Metareview:**

The paper studies the problem of enhancing neural network robustness from a dynamic system perspective. To this end, the authors proposed a nonautonomous neural ordinary differential equation (ASODE) that makes clean instances be their asymptotically stable equilibrium points. In this way, the asymptotic stability will reduce the adversarial noise to bring the nearby adversarial examples close to the clean instance. The empirical studies show that the proposed method can defend against existing attacks and outperform SOTA defense methods. Most reviewers rated this work positively and agreed that the proposed method is interesting, technically sound, and theoretically grounded. In the original reviews, they also raised several concerns including missing ablation study and sensitivity analysis, missing references, and some presentation issues. The authors properly addressed these concerns in the rebuttal and after the discussion, some reviewers increased their scores and the majority leaned towards acceptance. Overall, given the general support from the reviewers and the revised version of the paper, I recommend accepting the paper.

**Award:**

No

---

### Decision · Program_Chairs · 2022-09-14

Accept